# Adversarial Robustness of Implicit Neural Representation-Based Classifiers

**Jayoung Kim**[1]   **Kookjin Lee**[2]   **Noseong Park**[1]   **Sanghyun Hong**[3]

## Abstract

Implicit neural representations (INRs) encode data as continuous coordinate-based functions parameterized by neural networks, shifting downstream tasks such as image recognition to operate on functional rather than discrete representations. Despite their increasing adoption, the adversarial robustness of INR-based classification pipelines remain largely underexplored. In this work, we present the first systematic study of adversarial robustness in INR-based classifiers. A key challenge is that generating an INR requires *training* a neural network for each input sample, resulting in an optimization-in-the-loop forward pass that renders standard gradient-based attacks computationally prohibitive. To address this, we design surrogate models that amortizes the INR-generation process, serving as a practical proxy for attacking INR-based classifiers. We also develop speed-up techniques that substantially reduce the training cost of the surrogate. We show that in contrast to recent work, INR-based classifiers are vulnerable: under adversarial input perturbations, classification accuracy substantially degrades. Moreover, existing countermeasures designed to operate on discrete representations offer limited protection.

## 1. Introduction

Implicit neural representations (INRs) have emerged as an alternative representation for visual data, modeling signals as continuous coordinate-based functions parameterized by neural networks (Sitzmann et al., 2020; Tancik et al., 2020; Mildenhall et al., 2021; Ramasinghe & Lucey, 2022). This paradigm has attractive growing research interest due to its expressiveness and flexibility, and recent work has begun to explore the use of INRs beyond signal reconstruction, in-

cluding their incorporation into classification pipelines (Xu et al., 2022; Zhou et al., 2023b;a; De Luigi et al., 2023; Navon et al., 2023; Kalogeropoulos et al., 2024; Papa et al., 2024; Gielisse & van Gemert, 2025).

Despite their growing interest, the adversarial robustness of INR-based classifiers remain poorly understood. Prior work on adversarial robustness has largely focused on standard classifiers operating directly in pixel space (Madry et al., 2018; Goodfellow et al., 2014; Szegedy et al., 2013). A recent exploratory work has suggested that INR-based classifiers may exhibit improved robustness when attacks require differentiating through the INR-generation process, effectively *obfuscating* gradients via optimization (Shor et al., 2025). However, these claims have not been systematically evaluated in end-to-end classification settings. It is also unclear whether robustness insights and benefits observed for conventional models carry over to INR-based classifiers.

In this work, we systematically audit the adversarial robustness of INR-based classifiers. Rather than proposing stronger attacks, our goal is to answer this research question: *How robust are INR-based classifiers when evaluated under realistic threat models? (cf. Figure 1)* Addressing this question presents unique challenges, as INRs are generated via instance-specific optimization that requires training of a neural network for each input. This optimization-in-the-loop structure renders standard first-order evaluation methods, such as PGD attacks (Croce & Hein, 2020; Madry et al., 2018), either not applicable or computationally prohibitive.

To address this challenge, we present a surrogate-based robustness auditing framework for INR-based classifiers. Our surrogate modeling approach draws inspiration from non-intrusive gradient estimation methods (Arisaka & Li, 2024) that approximate gradients of otherwise non-differentiable processes via adaptive surrogate models. By approximating the INR-generation process, our framework enables the application of standard first-order evaluation methods despite the non-differentiability of INR fitting. It also systematically examines multiple attack surfaces: those where robustness can be audited via first-order methods, allowing for comprehensive evaluation across pipeline designs.

Our evaluation shows that INR-based classifiers are not robust to adversarial examples, contrary to recent optimistic findings (Rusu et al., 2022; Shor et al., 2025). Across classi-

[1]Korea Advanced Institute of Science & Technology, Daejeon, South Korea [2]Arizona State University, Tempe AZ, USA [3]Oregon State University, Corvallis OR, USA. Correspondence to: Sanghyun Hong <sanghyun.hong@oregonstate.edu>.

*Proceedings of the 43rd International Conference on Machine Learning*, Seoul, South Korea. PMLR 306, 2026. Copyright 2026 by the author(s).

fier designs, adversarial perturbations substantially degrade classification accuracy, suggesting representation-specific vulnerabilities rather than conventional pixel-space weaknesses. We also observe that increasing INR expressivity increases this vulnerability, making downstream classifiers more susceptible to attack. Moreover, we find that existing defenses and their adaptations offer limited robustness gains for INR-based classifiers, often at substantial costs in accuracy and computational overhead.

In summary, our contributions are as follows:

- We conduct a systematic study of adversarial robustness in INR-based classification pipelines, evaluating six downstream models across three pipeline designs on three image and a 3D point-cloud benchmarks.

- We introduce a surrogate-based auditing framework that addresses the non-differentiability of INR fitting by approximating the INR-generation process, enabling standard first-order robustness evaluations despite the optimization-in-the-loop nature of INR generation.

- We show that INR-based classifiers are not inherently robust, contrary to prior optimistic findings. Across benchmarks, our attacks substantially degrade classification accuracy, suggesting representation-specific vulnerabilities that grows with INR expressivity.

- We evaluate existing defenses and their INR-specific adaptations. We find that they provide robustness gains, often at substantial accuracy and computational cost.

## 2. Background and Related Work

Our work sits at the intersection of implicit neural representations and adversarial robustness. We provide an overview of the necessary background to contextualize our study.

### 2.1. Implicit Neural Representations

An implicit neural representation (INR) encodes a signal as a continuous function $f_\theta : \mathbb{R}^m \to \mathbb{R}^n$ parameterized by a neural network, where $\theta \in \mathbb{R}^d$ denotes its weights and biases. The neural network is typically chosen to be a small multilayer perceptron (MLP). For images, the input corresponds to 2D pixel coordinates and the output represents RGB values. Given an image $x$, fitting an INR requires solving an *instance-specific optimization* problem (i.e., training of a neural network):

$$\theta^*(x) = \arg\min_\theta \|f_\theta(\mathcal{G}) - x\|^2, \qquad (1)$$

where $\mathcal{G}$ denotes the set of pixel grid coordinates. There exist several variants of INRs that differ in architecture and activation functions (Mildenhall et al., 2021; Tancik et al., 2020; Saragadam et al., 2023). In this work, we focus on SIREN (Sitzmann et al., 2020) as a representative one, as it

is widely adopted and known for its strong expressivity for signals with high-frequency components.

**INR-based classifiers.** INRs underpin remarkable results in novel-view synthesis (NeRF) (Mildenhall et al., 2021), super-resolution (Aiyetigbo et al., 2025), 3-D reconstruction (Park et al., 2019), medical imaging (Shen et al., 2022) and video compression (Chen et al., 2021). We focus on image classification, as it provides a simple yet representative setting for systematic auditing of adversarial robustness, consistent with prior work in this area.

Classification pipelines employ INRs as follows. An INR is first fitted to obtain $\theta^*(x)$ via Equation (1) for each image with a small number of training iterations (gradient-descent steps). The fitted INR parameters $\theta^*$ can then be processed in several ways: they may be fed directly into a classifier (Navon et al., 2023; Zhou et al., 2023a; Kalogeropoulos et al., 2024), encoded into a compact embedding via weight-space encoders such as Inr2Vec (De Luigi et al., 2023) or Neural Functional Transformers (NFT) (Zhou et al., 2023b), or jointly optimized with the classifier in an end-to-end manner (Gielisse & van Gemert, 2025). Despite differences in pipeline design, these approaches share a common structure in which the INR acts as an intermediate representation between input data and final classification. Accordingly, we study six representative INR-based classifiers, which can be grouped into three categories based on their pipeline design:

*(1) Direct classification.* This pipeline directly processes the INR parameters $\theta^*$ via a classifier $C$ to predict the label:

$$\hat{y} = C(\theta^*(x)).$$

This category includes DWS (Navon et al., 2023), NFN (Zhou et al., 2023a), and ScaleGMN (Kalogeropoulos et al., 2024), which design classifiers that respect permutation or scale symmetries in the INR weight space.

*(2) Encoder-based classification.* In this pipeline, the INR parameters $\theta^*$ are first encoded into a compact embedding $h$ via an encoder $E$, which is then passed to a classifier:

$$\hat{y} = C(h), \quad h = E(\theta^*(x)).$$

Inr2Vec (De Luigi et al., 2023) and NFT (Zhou et al., 2023b) fall into this category, where the encoder learns to map INR weights into a fixed-dimensional latent space.

*(3) End-to-end classification.* MWT (Gielisse & van Gemert, 2025) adopts a different design by jointly optimizing the INR fitting process and the classifier in an end-to-end manner using a meta-learned initialization. By enabling INR fitting in only a small number of gradient steps, this approach makes it computationally feasible to differentiate through the fitting process, thereby allowing standard *first-order* robustness evaluation methods to be applied. See Appendix B.4 for more details.

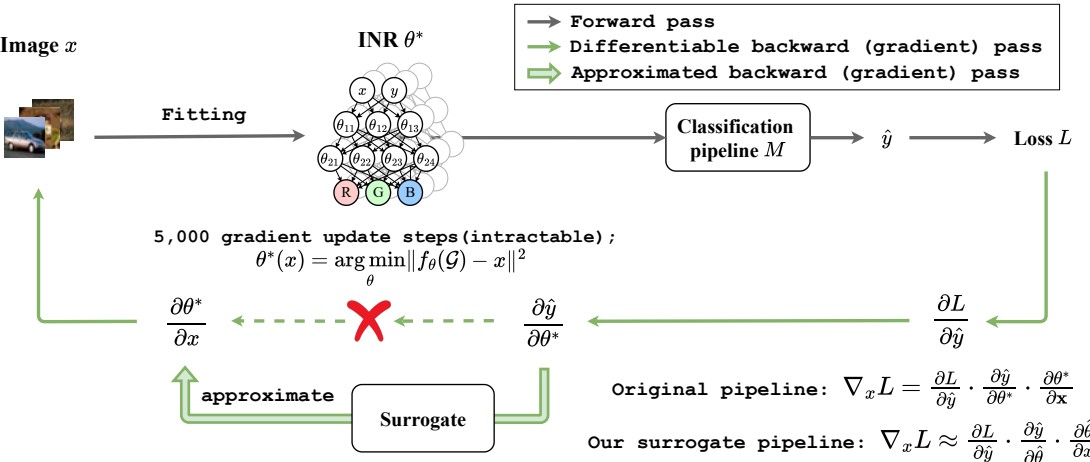

*Figure 1.* **Surrogate-based auditing of adversarial robustness in INR-based classifiers.** Direct attacks on INR-based classifiers are challenging because they require implicitly solving instance-specific INR fitting. Our surrogate-based attacks approximates this INR generation and enables first-order attacks. In this way, we expose a function-space vulnerability beyond conventional input-space ones.

### 2.2. Robustness Auditing and Evaluation

Prior work has shown that neural networks are vulnerable to *adversarial examples* (Szegedy et al., 2013; Goodfellow et al., 2014; Papernot et al., 2017; Carlini & Wagner, 2017; Madry et al., 2018). An adversary who wants to maximize a model's error on specific test-time samples can introduce human-imperceptible perturbations to them.

**Projected Gradient Descent (PGD)** is a standard first-order method for generating adversarial examples. PGD iteratively perturbs an input to maximize the loss of a target model under a norm-bounded constraint. Given a clean input $x$ with label $y$ and a perturbation budget $\varepsilon$, it solves:

$$\max_{x'} L(x', y), \text{ s.t. } \|x' - x\|_\infty \leq \varepsilon$$

through the iterative update:

$$x^{t+1} = \Pi_\varepsilon(x^t + \alpha \cdot \text{sign}(\nabla_x L(x^t, y))),$$

where $\alpha$ denotes the step size and $\Pi_\varepsilon$ projects the perturbed input back onto the $\ell_\infty$ ball of radius $\varepsilon$ centered at the original input $x$. PGD is commonly used in the *white-box* setting for robustness auditing, where the adversary has full access to the model parameters and the loss function.

**Surrogate-based attacks.** When direct access to gradients is unavailable or computationally prohibitive, adversaries often resort to surrogate models to approximate the target pipeline and craft adversarial examples. Prior work has shown that adversarial examples generated on a surrogate model can transfer to other models (Athalye et al., 2018; Tramèr et al., 2017; Inkawhich et al., 2020), enabling practical attacks in black-box or partially observable settings. This transferability has motivated surrogate modeling as a general strategy for probing the robustness of non-differentiable or costly-to-differentiate systems.

**Defenses.** An extensive body of work (Kurakin et al., 2018; Xu et al., 2017; Song et al., 2017; Liao et al., 2018; Lecuyer et al., 2019) has proposed defenses against adversarial examples. Among these, randomized smoothing offers a general framework for robustness by constructing smoothed classifiers, and in some variants (Cohen et al., 2019), provides formal robustness guarantees. Within this framework, different training strategies can be used to obtain a robust base classifier. One common approach is to employ *adversarial training* (Szegedy et al., 2013; Madry et al., 2018), where adversarially perturbed versions of clean training samples are generated and used to update the model, improving its empirical robustness and strengthening the resulting smoothed classifier. Another line of work focuses on *denoised smoothing* (Salman et al., 2020; Carlini et al., 2023), which applies a denoising transformation to the input prior to smoothing in order to reduce the impact of adversarial perturbations.

Our work introduces a novel surrogate modeling approach that approximates the INR generation process (§3), enabling systematic robustness auditing of INR-based classifiers (§4). We further evaluate the effectiveness of the two front-runner defenses under this auditing framework (§4.4).

## 3. Our Auditing Methodology

We introduce our approach for auditing the adversarial robustness of INR-based classifiers. We enable a systematic evaluation across different pipeline designs by addressing a key challenge—the non-differentiability of the INR generation process. To this end, we develop a surrogate modeling method that accurately approximates backward gradients, enabling the application of standard first-order methods. Figure 1 illustrates the gradient bottleneck in INR-based classification and how our surrogate approximates it.

## 3.1. Threat Model

We consider an adversary whose goal is to induce indiscriminate misclassification in an INR-based classifier. Following standard assumptions in prior work on adversarial robustness (Madry et al., 2018), we adopt a *white-box* attacker who has full knowledge of the INR and classifier architectures, their trained parameters, and the algorithms used to construct and process INRs. However, we assume that the adversary does not have access to the specific random initialization used during INR fitting, a practical assumption since such initializations are typically not disclosed.

A white-box adversary may operate over three attack surfaces: the input image space, the INR parameter space, and the embedding space. While INR- and embedding-space attacks are less likely to arise in practice, yet they provide valuable insights into representation-specific vulnerabilities within INR-based classifiers. Input-space attacks, despite more challenging due to the non-differentiable INR-generation process, are critical for evaluating practical, end-to-end robustness.

## 3.2. Attacking on INR- and Embedding-Space

A white-box adversary may also directly perturb intermediate representations within INR-based classification pipelines. Let $M$ denote the downstream model that maps fitted INR parameters $\theta^*$ to predictions. Since $M$ is fully differentiable, gradients with respect to these intermediate representations can be computed directly, allowing straightforward adaptations of standard PGD attacks.

**INR-Space Attack.** A practical challenge in function-space attacks is to perturb INR parameters while keeping the induced changes in the rendered image bound. Let $\theta^* = \theta^*(x)$ denote the INR parameters fitted to a clean image $x$. Given the label $y$, we compute a perturbation $\Delta\theta$ that maximizes the classification loss while constraining the induced change in image space as follows:

$$\max_{\Delta\theta} \quad L(M(\theta^* + \Delta\theta), y)$$
$$\text{s.t.} \quad \|f_{\theta^* + \Delta\theta}(\mathcal{G}) - f_{\theta^*}(\mathcal{G})\|_\infty \leq \varepsilon.$$

We run PGD attack in weight space and project perturbations to satisfy the input-space constraint after each step (see Appendix A for details).

**Embedding-Space Attack.** For encoder-based pipelines, we directly perturb the embedding $h = E(\theta^*)$. Because embeddings are unbounded, absolute perturbation budgets are less meaningful; we instead normalize perturbations by the empirical scale of the embedding distribution. Specifically, we constrain

$$\|h' - h\|_\infty \leq \text{std}(H) \cdot \varepsilon_{\text{rel}},$$

where $H = \{E(\theta^*(x_i))\}_i$ denotes the dataset embeddings, $\text{std}(H)$ is its empirical standard deviation, and $\varepsilon_{\text{rel}}$ is a calibrated relative perturbation budget. This attack provides a *diagnostic* upper bound on vulnerability but is only applicable to encoder-based pipelines and requires access to precomputed embeddings (see Appendix A).

## 3.3. Attacking INR-based Classifiers

In previous sections, we extend existing first-order adversarial attacks to the INR setting. However, unlike models that operate directly on pixel-space inputs, INR-based classifiers require fitting an INR for each image. Input-space attacks thus must differentiate through a *training* procedure, which makes direct gradient computation prohibitively expensive. We present our approach for overcoming this challenge and enabling tractable first-order attacks in image space.

**Gradient bottlenecks in INR fitting.** We now describe the key challenge underlying input-space adversarial attacks on INRs. Given an image $x$, fitting an INR corresponds to solving the optimization problem in Equation (1), which involves thousands of gradient descent steps from an initialization $\theta_0$ to obtain $\theta^*$. Recall that $M$ denote the downstream model mapping $\theta^*$ to predictions, yielding:

$$\hat{y} = M(\theta^*(x)).$$

(1) $M = C$ for the classifiers operates directly on INRs $\theta^*$. (2) For encoder-based pipelines, $M = C \circ E$ first encodes the INR into an embedding before classification.

To craft adversarial examples using the first-order methods, the adversary must compute the input gradient $\nabla_x L(y, \hat{y})$, where $L$ is the cross-entropy loss. By the chain rule:

$$\nabla_x L = \frac{\partial L}{\partial \hat{y}} \cdot \frac{\partial \hat{y}}{\partial \theta^*} \cdot \frac{\partial \theta^*}{\partial x}.$$

Since classifier $C$ and/or encoder $E$ remain differentiable, the first two terms are differentiable and can be easily computed via automatic differentiation in standard deep-learning frameworks, e.g., PyTorch (Paszke et al., 2019). In contrast, the final term $\frac{\partial \theta^*}{\partial x}$ requires differentiation through the INR fitting procedure. Although this is in principle computable by unrolling the optimization, doing so requires storing intermediate states across thousands of gradient steps, making direct gradient computation intractable in practice.

**Surrogate gradients for INR attacks.** To overcome the computational costs of differentiating through INR fitting, we introduce surrogate models that approximate the gradients required for input-space first-order attacks. Our key insight is that exact gradients are *not necessary*: it is sufficient to approximate adversarially useful directions in input space.

**INR surrogates** $\hat{f}$ directly approximate $\frac{\partial \theta^*}{\partial x}$ by predicting $\hat{\theta} = \hat{f}(x) \approx \theta^*$. We consider three variants:

(1) **Hypernetwork:** We adopt the hypernetwork of (Sitzmann et al., 2020) which predicts SIREN weights using a convolution encoder and a hypernetwork decoder, trained with reconstruction and classification losses.

(2) **Naïve INR surrogate:** A simple MLP encoder trained to minimize the MSE between predicted and fitted INR weights, along with a classification loss.

(3) **FD INR surrogate:** Extends Naïve INR surrogate with finite-difference (FD) supervision. Given small perturbations to INR parameters $\Delta\theta$, we compute the induced changes in rendered images $\Delta x$. We then supervise the surrogate's Jacobian to satisfy $\frac{\partial \hat{\theta}}{\partial x} \cdot \Delta x \approx \Delta\theta$. This encourages the surrogate's Jacobian to locally approximate that of the true INR fitting, improving gradient estimation for PGD.

**Embedding surrogates** $\hat{g}$ approximate $\frac{\partial E(\theta^*)}{\partial x}$ directly by predicting $\hat{h} = \hat{g}(x) \approx E(\theta^*)$ bypassing both fitting and encoding. Because the classifier $C$ remains fully differentiable, this provides a more direct approximation to the classification loss. We consider two variants:

(1) **Naïve Embedding surrogate:** A MLP encoder trained to directly match the target embedding in latent space, along with a classification loss.

(2) **FD Embedding surrogate:** Extends Naïve Embedding surrogate with finite-difference supervision to align the surrogate's Jacobian with $\frac{\partial E(\theta^*)}{\partial x}$.

Loss functions and architectural details are provided in Appendix B.2. INR surrogates apply to all pipelines, while embedding surrogates are specific to encoder-based pipelines. After training, we perform PGD using the surrogate's gradients to generate adversarial examples, which are then evaluated through the same surrogate-based pipeline.

## 4. Empirical Evaluation

We now evaluate the robustness of INR-based classifiers.

### 4.1. Experimental Setup

We design our experimental setup to comprehensively evaluate the adversarial robustness across datasets, models, and attack strategies. Below we provide a summary of the setup; additional details, such as architectural configurations and hyperparameter choices, are included in Appendix B.

**Datasets.** We evaluate our auditing method on four benchmarks spanning 2D image and 3D point-cloud classification. For image recognition, we use three standard benchmarks: MNIST (LeCun et al., 2010), FashionMNIST (Xiao et al., 2017), and CIFAR-10 (Krizhevsky et al., 2009). These datasets are widely studied and increasingly used in recent work on INRs. To construct INRs from images, we use SIREN (Sitzmann et al., 2020) with the same architectural configuration as NFN (Zhou et al., 2023a). To assess whether our findings generalize beyond 2D images, we additionally evaluate on ShapeNet-10 (Chang et al., 2015), a 3D point-cloud classification benchmark, following the setup of Inr2Vec and NFN and the perturbation budget of Sun et al. (2021). See Appendix B.1 for more details.

**Models.** We comprehensively evaluate *six* downstream classifiers: DWS (Navon et al., 2023) employs an MLP architecture that enforces equivariance to permutation symmetries in INR parameters. Inr2Vec (De Luigi et al., 2023) encodes INRs into compact latent codes for classification. NFN (Zhou et al., 2023a) and NFT (Zhou et al., 2023b) incorporate permutation equivariance in the INR weight space when generating compact encodings. ScaleGMN (Kalogeropoulos et al., 2024) extends this framework by incorporating scale equivariance. MWT (Gielisse & van Gemert, 2025) leverages a meta-learned initialization strategy that enables end-to-end differentiable INR fitting.

**Attacks.** To compare the effectiveness of our methods, we run the following attacks:

- **Random noise:** A baseline that adds Gaussian noise to input images, serving as a sanity check.

- **Transfer-based attacks:** We train a convolutional neural network on explicit image representations and use it as a surrogate to craft adversarial examples, which are then *transferred* to the INR-based classifiers. We use ResNet-18 as the surrogate model.

- **Our attacks:** We run standard PGD using gradients obtained by backpropagating through our differentiable surrogates of the INR generation process. This makes first-order attacks applicable despite the non-differentiability of INR fitting. We train five types of surrogate models as described in §3.3: Hypernetwork, Naïve INR, FD INR, Naïve Embedding, and FD Embedding surrogates. INR surrogates are applicable to all pipelines, while embedding surrogates are only applicable to encoder-based pipelines, specifically Inr2Vec and NFT.

Following the standard practices in prior work on adversarial robustness (Madry et al., 2018), we use 20-step PGD, leveraging gradient obtained from trained surrogates. We set the bound to an $\ell_\infty$-norm of 0.3 for MNIST, 0.1 for Fashion-MNIST, and 0.03 for CIFAR-10. To assess robustness beyond PGD, we additionally evaluate our framework under a function-space Square Attack (Andriushchenko et al., 2020), DeepFool (Moosavi-Dezfooli et al., 2016), and a One-Pixel attack (Su et al., 2019). Their formulations and full results are provided in Appendix C.8.

*Table 1.* **Adversarial robustness of INR-based classifiers.** Attacks are bounded to $\ell_\infty$-norm 0.3 for MNIST, 0.1 for F-MNIST, and 0.03 for CIFAR-10. We report the classifier's clean accuracy (*Classifier*), and perturbed accuracy (%) under each attack method. All perturbed samples are evaluated on the *surrogate pipeline*. "Best Surrogate" denotes the strongest result among applicable surrogate methods, i.e., Hypernetwork, Naïve INR, FD INR, Naïve Embedding, and FD Embedding surrogates, for each classifier. "–" indicates the attack is not applicable to that model architecture.

| Dataset | Attack Method | Inr2Vec | DWS | NFN | NFT | ScaleGMN | MWT |
|---------|---------------|---------|-----|-----|-----|----------|-----|
| **MNIST** | *Classifier* | 88.99% | 75.48% | 90.76% | 98.36% | 94.75% | 98.50% |
| | Random | 78.21% | 26.53% | 39.04% | 77.54% | 79.44% | 77.60% |
| | Transfer | 89.37% | 75.81% | 90.35% | 98.20% | 94.47% | 59.20% |
| | Best Surrogate | **0.01%** | **0.01%** | **0.17%** | **0.00%** | **0.00%** | – |
| **F-MNIST** | *Classifier* | 78.67% | 74.63% | 79.27% | 83.79% | 82.78% | 89.50% |
| | Random | 77.07% | 68.33% | 74.77% | 80.92% | 81.05% | 89.40% |
| | Transfer | 78.26% | 74.47% | 78.85% | 83.83% | 82.21% | 88.10% |
| | Best Surrogate | **0.13%** | **15.15%** | **0.27%** | **0.78%** | **10.44%** | – |
| **CIFAR-10** | *Classifier* | 35.13% | 39.63% | 41.50% | 48.87% | 48.56% | 56.90% |
| | Random | 31.12% | 32.22% | 35.62% | 45.19% | 41.29% | 56.70% |
| | Transfer | 33.71% | 37.41% | 39.81% | 47.72% | 45.98% | 56.3% |
| | Best Surrogate | **0.46%** | **0.38%** | **0.15%** | **0.15%** | **0.25%** | – |

**Metrics.** We measure classification accuracy (Acc) on clean and adversarial examples, computed on the entire test-set.

### 4.2. Adversarial Robustness of INR-based Classifiers

We first evaluate how INR-based classifiers are vulnerable to adversarial examples (**RQ1**) and which surrogate methods are the most effective in auditing the robustness (**RQ2**).

**INR-based classifiers are resilient to trivial attacks.** Table 1 summarizes our findings. Transfer-based attacks prove largely ineffective across all pipelines. Adversarial perturbations crafted on ResNet-18 do not preserve their effect after passing through INR fitting. Random Gaussian noise reveals differences across pipeline designs. Direct classification pipelines such as DWS and NFN, which feed INR parameters directly to the classifier, are vulnerable under standard perturbation budgets, e.g., Acc drops by 49% and 52% at $\varepsilon = 0.3$ for DWS and NFN on MNIST. Encoder-based pipelines, i.e., Inr2Vec and NFT remain relatively stable, as the encoder learns to extract class-relevant features while ignoring noise-induced variations in INR parameters.

**Our attacks effectively approximate vulnerable gradient directions.** We next evaluate whether our approach produces stronger adversarial examples, enabling robustness auditing of INR-based classifiers against empirical worst-cases. As shown in Table 1, adversarial examples generated using our best surrogate reduce classifier accuracy to nearly zero. Table 2 compares individual surrogates. Naïve INR surrogates, trained without gradient direction supervision, produce substantially weaker attacks: Acc remains at 10–20% on DWS and NFT on MNIST. In contrast, FD INR surrogates, which incorporate finite-difference supervision to better align gradients with the true INR fitting process, reduce Acc to near

*Table 2.* **Comparison of surrogate methods.** We report accuracy (%) under each surrogate-based attack. Embedding surrogates only apply to encoder-based pipelines (Inr2Vec and NFT). Best results per pipeline are in **bold**. "–" indicates that a surrogate is not applicable. Complete results are provided in Appendix C.1.

| Dataset | Surrogate | Inr2Vec | DWS | NFN | NFT | ScaleGMN |
|---------|-----------|---------|-----|-----|-----|----------|
| **MNIST** | *Classifier* | 88.99% | 75.48% | 90.76% | 98.36% | 94.75% |
| | Hypernetwork | **0.01%** | 0.03% | **0.17%** | **0.00%** | 0.09% |
| | Naïve INR | 2.19% | 10.12% | 0.29% | 20.57% | 7.16% |
| | FD INR | 1.91% | **0.01%** | 0.40% | **0.00%** | **0.00%** |
| | Naïve Emb. | 0.69% | – | – | **0.00%** | – |
| | FD Emb. | 2.08% | – | – | **0.00%** | – |
| **F-MNIST** | *Classifier* | 78.67% | 74.63% | 79.27% | 83.79% | 82.78% |
| | Hypernetwork | **0.13%** | 15.51% | 0.76% | **0.78%** | 30.44% |
| | Naïve INR | 17.91% | 20.82% | 5.68% | 12.41% | 13.01% |
| | FD INR | 13.89% | **15.15%** | **0.27%** | 3.83% | **10.44%** |
| | Naïve Emb. | 21.63% | – | – | 9.40% | – |
| | FD Emb. | 27.20% | – | – | 21.53% | – |
| **CIFAR-10** | *Classifier* | 35.13% | 39.63% | 41.50% | 48.87% | 48.56% |
| | Hypernetwork | 0.48% | **0.38%** | 0.40% | **0.15%** | 1.07% |
| | Naïve INR | **0.46%** | 3.70% | 4.95% | 4.13% | 3.96% |
| | FD INR | 5.22% | 2.84% | **0.15%** | 0.19% | **0.25%** |
| | Naïve Emb. | 4.58% | – | – | 0.82% | – |
| | FD Emb. | 13.59% | – | – | 2.13% | – |

zero. To verify that this gain reflects faithful gradient approximation rather than a surrogate-specific artifact, we ablate the strength of the finite-difference supervision ($\lambda_{\text{der}}$) on MNIST: stronger supervision yields lower Jacobian loss—closer alignment to the true INR fitting gradient—and consistently stronger attacks, while surrogate clean accuracy remains stable (See Appendix C.2). Hypernetwork surrogates achieve comparable attack strength but require a more complex architecture and exhibit training instability. These results highlight that relying on straightforward approach alone risks obscuring vulnerabilities due to gradient masking (Athalye et al., 2018), giving the false impression of adversarial robustness.

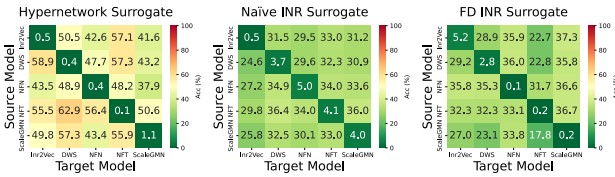

*Figure 2.* **Cross-pipeline transferability on CIFAR-10.** Each heatmap shows perturbed accuracy (%) when adversarial examples crafted using the source model's surrogate (row) are evaluated on the target model (column). Lower values indicate stronger attacks. Diagonal entries represent same-model attacks; off-diagonal entries represent transfer attacks.

**Our adversarial examples *transfer* across pipelines.** We further investigate whether adversarial examples crafted on one INR-based classification pipeline are effective against others. The results are summarized as a transferability heatmap in Figure 2. We observe moderate cross-pipeline transferability: adversarial examples generated on

one model degrade accuracy on other models, but transfer attacks are substantially weaker than same-model attacks. For instance, with the FD INR surrogate, same-model attacks reduce accuracy to 0.2–5.2%, while transfer attacks yield 17–37% accuracy. This gap indicates that our surrogate-based attacks exploit both shared vulnerabilities across INR-based pipelines and model-specific weaknesses. Among the three INR surrogates, Naïve INR shows the most consistent transfer (24–36%), while Hypernetwork exhibits the largest gap between same-model and transfer performance.

*Table 3.* **Generalization to 3D point-cloud classification.** We report clean accuracy and perturbed accuracy (%) under each surrogate-based attack on ShapeNet-10 ($\varepsilon = 0.05$, following Sun et al. (2021)). For each pipeline, the strongest attack result is in **bold**. Hypernetwork and embedding-space surrogates are not evaluated; surrogate clean accuracy is reported in Appendix C.1.

| Dataset | Surrogate | Inr2Vec | DWS | NFN | NFT | ScaleGMN |
|---|---|---|---|---|---|---|
| **ShapeNet-10** | *Classifier* | 92.7% | 76.4% | 87.4% | 92.9% | 88.8% |
| | Naïve INR | **53.9%** | 44.6% | **52.2%** | **37.4%** | **50.5%** |
| | FD INR | 55.9% | **42.9%** | 54.0% | 51.4% | 60.7% |

**Generalization to 3D data.** To assess whether our findings extend beyond 2D images, we evaluate our auditing framework on ShapeNet-10, a 3D point-cloud classification benchmark. As shown in Table 3, surrogate-based attacks consistently reduce accuracy across all evaluated classifiers (e.g., Inr2Vec $92.7\% \rightarrow 53.9\%$, NFT $92.9\% \rightarrow 37.4\%$). These results suggest that the robustness limitations of INR-based classifiers are not restricted to 2D image benchmarks, although the degradation is less severe than what we observe on 2D data. This difference is consistent with the greater difficulty of fitting INRs on 3D point clouds, as reflected in surrogate clean accuracy (cf. Table 10 in Appendix C.1).

A complementary analysis where adversarial examples are also refit through the classifier pipeline under a known initialization assumption is reported in Appendix D.

### 4.3. Factors Influencing the Adversarial Robustness

We next investigate the factors that most strongly influence adversarial robustness **(RQ3)**. We organize our analysis into three dimensions: (1) INR architecture, which determines how input signals are encoded into neural network weights; (2) classifier design, which determines how INR weights are processes for classification; and (3) attack configurations, which governs the strength and characteristics of adversarial perturbation. Due to space constraints, we report results on CIFAR-10; full results across all datasets are in Appendix C.

**Impact of INR architectures.** Figure 3 shows how the INR architecture affects robustness against FD INR surrogate on CIFAR-10. We vary three components from the default configuration (depth = 3, width = 32, $\omega_0 = 30$, 1.3K parameters): the number of hidden units per layer (width), the number of layers (depth), and frequency scale $\omega_0$, which de-

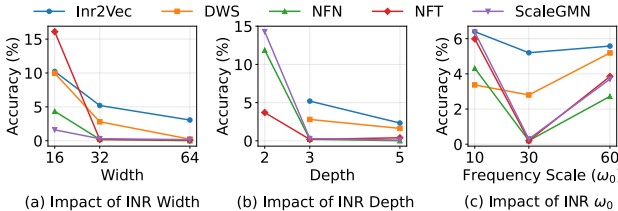

*Figure 3.* **Impact of INR architecture on adversarial robustness.** We vary (a) width (hidden units per layer), (b) depth (total number of layers), and (c) frequency scale $\omega_0$ of the SIREN architecture and measure accuracy under FD INR surrogate attack on CIFAR-10. Lower accuracy indicates higher vulnerability. Inr2Vec and DWS are omitted at depth=2 due to the architectural constraint that require a minimum number of layers.

*Table 4.* **Impact of # INRs used to train classifiers on CIFAR-10 under the FD INR attack**. Higher accuracy indicates weaker attacks. "Eq." indicates whether the model has built-in equivariance.

| Model | Eq. | Classifier Acc | | | Surrogate Acc | | | Perturbed Acc | | |
|---|---|---|---|---|---|---|---|---|---|---|
| | | 1 | 3 | 5 | 1 | 3 | 5 | 1 | 3 | 5 |
| Inr2Vec | ✗ | 35.1% | 41.21% | 32.5% | 51.0% | 41.3% | 42.0% | 5.2% | 5.9% | 8.8% |
| DWS | ✓ | 39.6% | 31.6% | 41.1% | 49.4% | 51.8% | 47.9% | 2.8% | 1.7% | 2.2% |
| NFN | ✓ | 41.5% | 45.0% | 45.7% | 52.0% | 51.1% | 54.1% | 0.2% | 1.1% | 4.4% |
| NFT | ✓ | 48.9% | 56.1% | 58.1% | 52.6% | 43.6& | 46.3% | 0.2% | 4.0% | 3.7% |
| ScaleGMN | ✓ | 48.6% | 51.3% | 52.1% | 49.8% | 51.0% | 48.9% | 0.3% | 1.3% | 1.9% |

termines how rapidly SIREN's activations oscillate—higher values enable the INR to represent finer, higher-frequency details. We find that deeper and wider architectures increase vulnerability, which aligns with model capacity (and expressivity): increasing width from 16 to 64 increases from 0.3K to 4.5K, and all configurations with higher capacity exhibit greater vulnerability. The effect of $\omega_0$ is non-monotonic: all pipelines exhibit highest vulnerability at $\omega_0 = 30$.

**Impact of classifier design.** We examine three aspects of pipeline design: the number of INRs fitted per image, equivariance properties, and embedding dimensionality. Table 4 shows the impact of fitting multiple INRs per image with different random initialization on CIFAR-10. Classifier accuracy generally improves with more INRs, as the model benefits from richer representations. However, surrogate clean accuracy tends to decrease, since the surrogate should approximate an one-to-many mapping from images to multiple INRs. For equivariant classifiers like NFN and ScaleGMN, surrogate clean accuracy is retained or even improves, yet attacks still become weaker—e.g,. NFN Acc increases from 0.2% to 4.4%. This behavior arises because equivariant classifiers are invariant to specific weight-space variations that our non-equivariant surrogates treat as semantically meaningful. As more INRs are fitted per image, the surrogate encounters more diverse weight configurations for the same underlying signal, amplifying this mismatch.

Table 5 summarizes our results across different embedding

*Table 5.* **Impact of embedding dimension on CIFAR-10.** Perturbed accuracy (%) under each attack; default dimension is 256.

| Surrogate | Inr2Vec | | | | NFT | | | |
|---|---|---|---|---|---|---|---|---|
| | 64 | 128 | 256 | 512 | 64 | 128 | 256 | 512 |
| *Classifier* | 34.71% | 38.50% | 35.13% | 40.16% | 47.37% | 48.35% | 48.16% | 48.16% |
| Hypernetwork | 0.68% | 0.57% | 0.48% | 0.63% | 0.04% | 0.18% | 0.15% | 0.63% |
| Naïve INR | 2.51% | 1.90% | 0.46% | 1.97% | 3.34% | 4.62% | 4.13% | 7.80% |
| FD INR | 4.62% | 6.00% | 5.22% | 7.91% | 3.96% | 4.03% | 0.19% | 7.08% |
| Naïve Emb | 0.67% | 0.92% | 4.58% | 0.92% | 0.88% | 0.83% | 0.82% | 4.63% |
| FD Emb | 3.16% | 2.78% | 13.61% | 0.85% | 0.86% | 1.30% | 2.13% | 5.09% |

dimensions. While larger embedding dimensions generally improve classifier accuracy for Inr2Vec, NFT shows relatively stable performance across dimensions. Their effect on adversarial robustness varies across configurations (see Appendix C.4).

**Impact of attack configurations.** The sensitivities of our framework to PGD steps, perturbation bound, and $\ell_p$ norm follow patterns established in prior adversarial-robustness literature; full results are reported in Appendix C.6.

**Conventional input-space vulnerabilities vs. a new function-space weakness.** We compare standard input-space attacks with function-space ones that directly perturb INR parameters or their embeddings. This comparison is particularly important because it identifies whether the observed vulnerabilities resemble conventional input-space adversarial examples or represent a new, representation-specific weakness. We test both attack classes on six pipelines and three datasets. We ensure that input-space and function-/embedding-space attacks operate under comparable perturbation budgets (see Appendix C.7 for details).

*Table 6.* **Attack surface comparison across pipelines.** We report accuracy (%) for input-, function-, and embedding-space attacks.

| Dataset | Attack Surface | Inr2Vec | DWS | NFN | NFT | ScaleGMN |
|---|---|---|---|---|---|---|
| **MNIST** | *Classifier* | 88.99% | 75.48% | 90.76% | 98.36% | 94.75% |
| | Input | 0.01% | 0.01% | 0.17% | 0.00% | 0.00% |
| | Function | 5.09% | 47.74% | 25.80% | 16.18% | 4.39% |
| | Embedding | 0.00% | – | – | 0.00% | – |
| **F-MNIST** | *Classifier* | 78.67% | 74.31% | 79.84% | 83.79% | 80.98% |
| | Input | 0.13% | 15.15% | 0.27% | 0.78% | 10.44% |
| | Function | 46.68% | 68.39% | 64.03% | 55.97% | 48.41% |
| | Embedding | 8.72% | – | – | 0.00% | – |
| **CIFAR-10** | *Classifier* | 35.13% | 39.63% | 41.50% | 48.87% | 48.56% |
| | Input | 0.46% | 0.38% | 0.15% | 0.15% | 0.25% |
| | Function | 30.78% | 38.52% | 34.59% | 39.82% | 32.37% |
| | Embedding | 7.01% | – | – | 0.00% | – |

Table 6 summarizes the results. Input-space attacks using our surrogates are the most effective, substantially reducing accuracy across all pipelines. In contrast, function-space attacks that directly perturb the fitted INR weights lead to only modest degradation. This gap highlights the advantage of our surrogate-based approach: by approximating the INR fitting process, the surrogates enable gradient-based

optimization that finds more effective perturbations than direct weight-space attacks. For encoder-based pipelines, embedding-space attacks fall between the two, with NFT showing near zero accuracy while Inr2Vec has 7% accuracy.

## 4.4. Improving the Robustness of INR-based Classifiers

We lastly study countermeasures for improving the robustness of INR-based classifiers against adversarial examples (**RQ4**). We first evaluate whether existing defenses improve robustness, and then introduce INR-specific adaptations and test whether they can further enhance robustness. We additionally explore an INR-specific regularization of the INR fitting objective in Appendix C.9.

*Table 7.* **Diffusion denoising against FD INR attacks:** Classifier accuracy on clean inputs (Clean), denoised clean inputs (D.Clean), perturbed images (Pert.), and denoised perturbed images (D.Pert.).

| Dataset | Classifier | Clean | Pert. | D.Clean | D.Pert. |
|---|---|---|---|---|---|
| **MNIST** | Inr2Vec | 88.99% | 1.91% | 89.21% | 1.78% |
| | DWS | 75.48% | 0.01% | 76.16% | 0.01% |
| | NFN | 90.76% | 0.40% | 90.33% | 0.42% |
| | NFT | 98.36% | 0.00% | 98.32% | 0.00% |
| | ScaleGMN | 94.75% | 0.00% | 94.72% | 15.12% |
| **F-MNIST** | Inr2Vec | 78.67% | 13.89% | 78.75% | 14.27% |
| | DWS | 74.63% | 15.15% | 73.86% | 15.22% |
| | NFN | 79.27% | 0.27% | 78.97% | 14.22% |
| | NFT | 83.79% | 3.83% | 83.92% | 9.28% |
| | ScaleGMN | 82.78% | 10.44% | 82.56% | 10.32% |
| **CIFAR-10** | Inr2Vec | 35.13% | 5.22% | 35.02% | 5.30% |
| | DWS | 39.63% | 2.84% | 39.62% | 3.02% |
| | NFN | 41.50% | 0.15% | 41.55% | 2.95% |
| | NFT | 48.87% | 0.19% | 48.97% | 3.39% |
| | ScaleGMN | 48.56% | 0.25% | 47.46% | 4.84% |

**Setup.** We test a provable defense by (Carlini et al., 2023) that leverages off-the-shelf diffusion denoisers to remove noise from images (Denoising). We also adapt adversarial training (AT) (Madry et al., 2018) to INR-based classifiers, considering three attack surfaces: (1) function-space AT, which perturbs fitted INR weights during training (applicable to all except MWT); (2) embedding-space AT, which generates adversarial examples directly in the embedding space (applicable to encoder-based pipelines); and (3) end-to-end AT, which produces adversarial examples by differentiating through the INR fitting process (MWT only).

**Diffusion denoising offers limited protection.** Table 7 summarizes results under FD INR surrogate attacks. On CIFAR-10, perturbed accuracy increases by at most 3–5%. On F-MNIST, improvements are slightly larger, where NFN improves to 14.2% from 0.3%. However, denoising has little to no defensive effect on MNIST. We also evaluate denoising on clean images to verify that the denoiser does not degrade the images themselves; this is not observed in D.Clean. This indicate that the perturbations generated by our surrogate-based attacks are not effectively removed by the diffusion model's reverse process.

*Table 8.* **Effect of function-space adversarial training.** We report change in clean ($\Delta$Cln) and perturbed ($\Delta$Prt) accuracy in percentage points. Positive $\Delta$Prt indicates improved robustness.

| Dataset | Inr2Vec | | DWS | | NFN | | NFT | | ScaleGMN | |
|---|---|---|---|---|---|---|---|---|---|---|
| | $\Delta$Cln | $\Delta$Prt | $\Delta$Cln | $\Delta$Prt | $\Delta$Cln | $\Delta$Prt | $\Delta$Cln | $\Delta$Prt | $\Delta$Cln | $\Delta$Prt |
| MNIST | -3.0 | +35.6 | +0.8 | +16.1 | -1.5 | +55.3 | -6.2 | +69.8 | +0.1 | +72.0 |
| F-MNIST | -1.4 | +22.2 | +0.9 | +2.8 | +0.3 | +14.2 | -5.1 | +17.5 | -2.5 | +20.6 |
| CIFAR-10 | +0.7 | -4.0 | +0.6 | +0.8 | -0.1 | -7.6 | -24.2 | -15.7 | -12.9 | +0.8 |

**Adversarial training shows data-dependent effectiveness.**
Table 8 summarizes the results. AT substantially improves the robustness on MNIST, with NFN increasing perturbed accuracy from 25.8% to 81.1% and NFT from 16.2% to 86.0%, with modest drops in clean accuracy. On F-MNIST, gains are moderate (e.g., NFN 64.0% $\rightarrow$ 78.2%, ScaleGMN 48.4% $\rightarrow$ 69.0%). In contrast, on CIFAR-10, AT provides limited benefit or even degrades performance: NFT's perturbed accuracy drops from 39.8% to 24.1% with clean accuracy falling from 48.9% to 24.7%. Inr2Vec shows marginal change; perturbed accuracy moves from 30.8% to 26.8%. We also evaluate end-to-end AT on MWT and embedding-space AT on encoder-based pipelines; results are in Appendix C.9.

## 5. Conclusion

Our work is the first to systematically *audit* the robustness of INR-based classifiers. To this end, we introduce a novel approach that addresses a key challenge: standard first-order methods for crafting adversarial examples are not applicable due to the non-differentiability of the INR-generation process. We overcome this limitation via training surrogate models capable of accurately approximating backward gradients. Our results show that those classifiers should use INRs with care. The models we study are *not* robust to adversarial examples, with vulnerabilities consistent across pipeline designs and rooted in representation-specific weaknesses rather than conventional input-space attacks. Moreover, while existing defenses and their adaptations can slightly improve robustness, they incur substantial trade-offs in accuracy and computational overhead. We hope our work draws attention to this underexplored vulnerability and motivates future work on countermeasures.

## Acknowledgements

We thank anonymous reviewers for their valuable feedback. S. Hong was partially supported by the Google Faculty Research Award. K. Lee acknowledges support from the U.S. National Science Foundation under grant IIS 2338909. The work of N. Park and J. Kim was partly supported by the Institute for Information & Communications Technology Planning & Evaluation (IITP) grants funded by the Korean government (MSIT) (No. RS-2026-25526850, High-Efficiency Neural Networks for Artificial General Intelligence; No. RS-2024-00457882, AI Research Hub Project; No. RS-2025-25442149, LG AI STAR Talent Development Program for Leading Large-Scale Generative AI Models in the Physical AI Domain), and Samsung Research Funding & Incubation Center of Samsung Electronics under Project Number SRFC-IT2402-08.

## Impact Statement

Our work supports socially responsible deployment of INR-based models by providing systematic auditing mechanisms that expose vulnerabilities to adversarial examples and by proposing defense strategies with minimal computational overhead. These contributions lower barriers to robustness assessment and defense and encourage the ML community to more rigorously examine robustness claims. Ultimately, our findings aim to promote the development and deployment of more secure and trustworthy ML systems.

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

# A. Attack details.

Here we describe the details of our attacks on each attack surface: input-space, function-space, and embedding-space.

## A.1. Perturbation Budget Calibration

To ensure fair comparison across attack surfaces, we calibrate perturbation budgets as follows. For input-space attacks, we use $\ell_\infty$ bounds of $\varepsilon = 0.3$ for MNIST, $0.1$ for Fashion-MNIST, and $0.03$ for CIFAR-10. For function-space attacks. we perturb weights in parameter space while constraining the induces change in rendered images to the same $\ell_\infty$ bound.

For embedding-space attacks, since embeddings are unbounded and cannot be rendered to images, we calibrate based on relative perturbation magnitude. We normalize the input-space bound $\varepsilon$ by the dataset's pixel standard deviation $\sigma_{\text{data}}$, then apply the same relative perturbation to embeddings:

$$\|h' - h\|_\infty \leq \text{std}(H) \cdot \varepsilon_{\text{rel}},$$

where $H = \{E(\theta^*(x_i))\}_i$ denotes the dataset embeddings and $\text{std}(H)$ its empirical standard deviation. We set $\varepsilon_{\text{rel}} = 0.12$ for CIFAR-10, $0.42$ for F-MNIST, and $0.94$ for MNIST.

## A.2. Function-Space Attack

A practical challenge in function-space attacks is to perturb INR parameters while keeping the induced changes in the rendered image bound. Let $\theta^* \in \mathbb{R}^d$ denote the INR parameters fitted to a clean image $x$, and let $f_\theta$ represent the implicit function parameterized by $\theta$. We obtain the corresponding image by evaluating the implicit function on a fixed pixel grid $\mathcal{G}$, i.e., $x = f_{\theta^*}(\mathcal{G})$. Given the label $y$, we compute a perturbation $\Delta\theta$ that maximizes the classification loss while constraining the induced change in image space as follows:

$$\max_{\Delta\theta} \quad L(M(\theta^* + \Delta\theta), y)$$
$$\text{s.t.} \quad \|f_{\theta^* + \Delta\theta}(\mathcal{G}) - f_{\theta^*}(\mathcal{G})\|_\infty \leq \varepsilon$$

We run PDG attacks in weight space,

$$\theta^{t+1} = \theta^t + \alpha \cdot \text{sign}\left(\nabla_\theta L(M(\theta^t), y)\right),$$

followed by a projection step onto the feasible set. Because the constraint is defined in image space rather than parameter space, we enforce it by rescaling the weight perturbation:

$$\theta^{t+1} \leftarrow \theta^* + \Pi^\varepsilon(\theta^{t+1} - \theta^*),$$

where

$$\Pi^\varepsilon(\Delta\theta) := \Delta\theta \cdot \min\left(1, \frac{\varepsilon}{\|f_{\theta^* + \Delta\theta}(\mathcal{G}) - f_{\theta^*}(\mathcal{G})\|_\infty}\right).$$

This projection shrinks the weight-space perturbation just enough to ensure that the resulting image remains within the $\ell_\infty$-ball of the bound $\varepsilon$ around $x_0$.

For function-space attacks, following Shor et al. (2025), we project the classification gradient orthogonally to the constraint gradient to prevent gradient steps from increasing constraint violation. Let $g_{\text{cls}} = \nabla_\theta L$ denote the classification loss gradient and $g_{\text{img}}$ denote the gradient of the $\ell_\infty$ input-space violation with respect to $\theta$. We project $g_{\text{cls}}$ orthogonally to $g_{\text{img}}$:

$$g_{\text{orth}} = g_{\text{cls}} - \frac{\langle g_{\text{cls}}, g_{\text{img}}\rangle}{\|g_{\text{img}}\|^2} g_{\text{img}}.$$

After each PGD step using $g_{\text{orth}}$, we project the weight perturbation to satisfy the input-space constraint by iteratively rescaling:

$$\theta^{t+1} \leftarrow \theta^* + (\theta^{t+1} - \theta^*) \cdot \min\left(1, \frac{0.9 \cdot \varepsilon}{\|f_{\theta^{t+1}}(\mathcal{G}) - f_{\theta^*}(\mathcal{G})\|_\infty}\right),$$

where the factor $0.9$ accounts for the nonlinearity of INR rendering. This is repeated until the constraint is satisfied.

## A.3. Embedding-Space Attack

For encoder-based pipelines, we directly perturb the embedding $h = E(\theta^*)$. Because embeddings are unbounded, absolute perturbation budgets are less meaningful; we instead normalize perturbations by the empirical scale of the embedding distribution. Specifically, we constrain

$$\|h' - h\|_\infty \leq \text{std}(H) \cdot \varepsilon_{\text{std}},$$

where $\text{std}(H)$ denotes the dataset-level standard deviation of embeddings and $\varepsilon_{\text{std}}$ is a calibrated relative perturbation budget. Given a clean embedding $h$ with label $y$, we apply PGD updates:

$$h^{t+1} = \Pi^{\varepsilon_{\text{rel}}}(h^t + \alpha \cdot \text{sign}(\nabla_h L(C(h^t), y))),$$

where $\Pi^{\varepsilon_{\text{rel}}}$ projects onto the $\ell_\infty$ ball with radius $\text{std}(H) \cdot \varepsilon_{\text{rel}}$ This attack provides a *diagnostic* upper bound on vulnerability but is only applicable to encoder-based pipelines and requires access to precomputed embeddings.

# B. Detailed Experimental Setup

**Code Availability.** Our code is available at `https://github.com/Trustworthy-and-Responsible-AI-Lab/INR-Robustness`. We run our experiments on a machine equipped with a 24-core Intel® Xeon Platinum processor, 512GB of DRAM, and 4 Nvidia RTX A6000 GPUs, each with 48GB of VRAM.

## B.1. INR Configuration

We follow the SIREN configuration from NFN (Zhou et al., 2023a) and NFT (Zhou et al., 2023b). Each INR is a 3-layer MLP with hidden dimension 32 and frequency scale $\omega_0 = 30$. We fit INRs using Adam optimizer with learning rate $5 \times 10^{-4}$ for 5,000 steps. For the 3D evaluation, we follow the ShapeNet-10 setup used by Inr2Vec (De Luigi et al., 2023), a 10-class subset of ShapeNet (Chang et al., 2015). Each INR is a 3-layer MLP with input dimension 3, output dimension 1, hidden dimension 128, frequency scale $\omega_0 = 30$. See De Luigi et al. (2023) for the full INR fitting procedure.

## B.2. Surrogates

- **Hypernetwork.** We adopt the hypernetwork architecture and code from SIREN (Sitzmann et al., 2020), adapted to match our SIREN configuration.
- **INR Surrogates.** Both Naïve and FD surrogates use a simple MLP with 512 hidden units and 3 hidden layers. Input dimension is the flattened image size; output dimension matches the total number of INR parameters. We train with AdamW optimizer, selecting learning rate from $\{1 \times 10^{-4}, 5 \times 10^{-4}, 1 \times 10^{-5}, 5 \times 10^{-5}\}$. Loss weights are selected from $\lambda_{\text{rec}} \in \{1, 10, 100\}$ and $\lambda_{\text{cls}} \in \{0.1, 1.0\}$. For FD surrogates, we additionally tune $\lambda_{\text{der}} \in \{0.1, 1.0\}$ and finite-difference step size $\alpha \in \{10^{-4}, 10^{-3}\}$.
- **Embedding Surrogates.** Architecture is identical to that of INR surrogates, with output dimension matching to classifier's embedding dimension. We use the same hyperparameter search space.

## B.3. Defenses

- **Diffusion denoising.** We use Improved DDPM (Nichol & Dhariwal, 2021) for the denoising diffusion model. For CIFAR-10, we use the publicly available checkpoint. For MNIST and F-MNIST, we train the models using the same codebase. We evaluate three denoising configurations; DDIM with 10→1 steps, DDPM with 1 step, and DDPM with 2 steps, and report the best result for each model.
- **Function-space Adversarial Training.** For direct classification pipelines, i.e., DWS, NFN, and ScaleGMN, we train with the objective from clean and perturbed samples:

$$L = 0.5 \cdot L_{\text{adv}} + 0.5 \cdot L_{\text{clean}},$$

where $L_{\text{adv}}$ is computed on function-space perturbed samples crafted using the same PGD configuration as evaluation. For encoder-based pipelines, i.e., Inr2Vec and NFT, the encoder is not trained to predict labels and thus cannot be adversarially trained with PGD. We therefore train the encoder using its original objective, freeze it, and adversarially train only the classifier.

- **Embedding-space Adversarial Training.** This is applicable only to encoder-based classifiers, i.e., Inr2Vec and NFT. We train the encoder using its original objective and freeze it. During classifier training, we apply PGD to the encoder outputs and train with the same objective as function-space adversarial training.

## B.4. Classifiers

We evaluate six representative INR-based classifiers across three pipeline designs. Below we describe each in detail.

- **Direct classification.** These pipelines feed INR parameters directly to a classifier without intermediate encoding.
  - **DWS (Navon et al., 2023)** processes INR weights using an MLP architecture that enforces equivariance to neuron permutation symmetries.
  - **NFN (Zhou et al., 2023a)** provides an alternative framework for permutation equivariant neural functionals. They introduce NF-layer operating on weight-space features under two settings: hidden neuron permutation (HNP) and full neuron permutation (NP), NP-layers are more parameter-efficient, while HNP is typically more appropriate for classification.
  - **ScaleGMN (Kalogeropoulos et al., 2024)** extends equivariance beyond permutations to scaling symmetries arising from activation functions, e.g., $\sigma(ax) = a\sigma(x)$ for ReLU with $a > 0$. It represents INRs as graphs and processes them via scale-equivariant message passing, handling various activations by adapting to their specific symmetries.

- **Encoder-based classification.** This pipelines first encodes INR parameters into a compact embedding for classification.
  - **Inr2Vec (De Luigi et al., 2023)** learns a mapping from INR weights to a fixed-dimensional latent code. This model uses a feature encoder for each MLP node followed by max-pooling, providing a simple baseline without explicit permutation equivariance.
  - **NFT (Zhou et al., 2023b)** leverages attention to define permutation equivariant weight-space layers. Its INR2ARRAY method maps INR weights to a spatially structured latent array $z \in \mathbb{R}^{M \times d}$ using an SNP-invariant encoder, where each vector encodes a spatial patch. A Transformer classification head then processes these latent arrays.

- **End-to-End classification.**
  - **MWT (Gielisse & van Gemert, 2025)** makes INR fitting differentiable via meta-learning. It learns both a shared SIREN initialization $\theta_0$ and per-parameter learning rates $\alpha$ fir $k$ inner-loop steps. Since $\theta_{k+1} = \theta_k - \alpha \nabla_\theta L_{\text{inner}}$ is differentiable, the classification loss can influence INR structure through back-propagation. Using few updates steps ($k = 4$–$6$) enables computational efficiency without requiring explicit weight-space equivariance.

We train all classifiers following their original papers. For hyperparameters not explicitly specified, we use the default values from official code releases.

# C. Additional Experimental Results

## C.1. Full Attack Comparison

*Table 9.* **Full comparison of attack methods across datasets.** We report surrogate clean accuracy (Clean) and perturbed accuracy (Pert.) under each attack. "–" indicates the attack is not applicable.

| Dataset | Attack | Inr2Vec | | DWS | | NFN | | NFT | | ScaleGMN | | MWT | |
|---|---|---|---|---|---|---|---|---|---|---|---|---|---|
| | | Clean | Pert. | Clean | Pert. | Clean | Pert. | Clean | Pert. | Clean | Pert. | Clean | Pert. |
| MNIST | *Classifier* | 88.99% | | 75.48% | | 90.76% | | 98.36% | | 94.75% | | 98.50% | |
| | Random | 88.60% | 79.09% | 75.48% | 26.53% | 90.76% | 39.04% | 98.36% | 77.54% | 94.75% | 79.44% | 98.50% | 77.60% |
| | Transfer | 88.60% | 88.71% | 75.48% | 75.81% | 90.76% | 90.35% | 98.36% | 98.20% | 94.75% | 94.47% | 98.50% | 59.20% |
| | Hypernetwork | 82.62% | **0.01%** | 29.84% | 0.03% | 81.11% | **0.17%** | 98.89% | **0.00%** | 85.19% | 0.09% | – | – |
| | Naïve INR | 95.57% | 2.19% | 97.17% | 10.12% | 95.27% | 0.29% | 97.65% | 20.57% | 95.92% | 7.16% | – | – |
| | FD INR | 96.01% | 1.91% | 96.36% | **0.01%** | 97.85% | 0.40% | 97.51% | **0.00%** | 96.15% | **0.00%** | – | – |
| | Naïve Emb. | 97.53% | 0.69% | – | – | – | – | 98.49% | **0.00%** | – | – | – | – |
| | FD Emb. | 95.98% | 2.08% | – | – | – | – | 97.13% | **0.00%** | – | – | – | – |
| F-MNIST | *Classifier* | 78.67% | | 74.63% | | 79.27% | | 83.79% | | 82.78% | | 89.50% | |
| | Random | 78.31% | 76.60% | 74.63% | 68.33% | 79.27% | 74.77% | 83.79% | 80.92% | 82.78% | 81.05% | 89.50% | 89.40% |
| | Transfer | 78.31% | 78.29% | 74.63% | 74.47% | 79.27% | 78.85% | 83.79% | 83.83% | 82.78% | 82.21% | 89.50% | 88.10% |
| | Hypernetwork | 82.24% | **0.13%** | 25.74% | 15.51% | 86.31% | 0.76% | 91.69% | **0.78%** | 52.54% | 30.44% | – | – |
| | Naïve INR | 87.84% | 17.91% | 88.73% | 20.82% | 88.41% | 5.68% | 88.37% | 12.41% | 88.59% | 13.01% | – | – |
| | FD INR | 87.21% | 13.89% | 88.61% | **15.15%** | 86.26% | **0.27%** | 88.82% | 3.83% | 82.23% | **10.44%** | – | – |
| | Naïve Emb. | 88.66% | 21.63% | – | – | – | – | 84.51% | 9.40% | – | – | – | – |
| | FD Emb. | 84.93% | 27.20% | – | – | – | – | 86.51% | 21.53% | – | – | – | – |
| CIFAR-10 | *Classifier* | 35.96% | | 39.63 | | 41.50% | | 48.87% | | 48.56% | | 56.90% | |
| | Random | 35.13% | 30.45% | 39.63% | 32.22% | 41.50% | 35.62% | 48.87% | 45.19% | 48.56% | 41.29% | 56.90% | 56.70% |
| | Transfer | 35.96% | 33.60% | 39.63% | 37.41% | 41.50% | 39.81% | 48.87% | 47.72% | 48.56% | 45.98% | 56.90% | 56.30% |
| | Hypernetwork | 45.21% | 0.48% | 20.42% | **0.38%** | 51.18% | 0.40% | 68.67% | **0.15%** | 41.87% | 1.07% | – | – |
| | Naïve INR | 49.51% | **0.46%** | 50.32% | 3.70% | 51.72% | 4.95% | 52.07% | 4.13% | 51.62% | 3.96% | – | – |
| | FD INR | 51.02% | 5.22% | 49.37% | 2.84% | 51.96% | **0.15%** | 52.60% | 0.19% | 49.78% | **0.25%** | – | – |
| | Naïve Emb. | 51.58% | 4.58% | – | – | – | – | 47.44% | 0.82% | – | – | – | – |
| | FD Emb. | 43.83% | 13.59% | – | – | – | – | 47.63% | 2.13% | – | – | – | – |

Table 9 summarizes the full attack results across all datasets, including surrogate accuracy. Surrogate-based attacks consistently outperform random Gaussian noise and transfer baselines, reducing accuracy to near zero in most cases. Higher surrogate accuracy generally correlates with stronger attacks, with the relationship varying across classifiers and datasets.

*Table 10.* **Surrogate clean and perturbed accuracy on ShapeNet-10.** Clean accuracy (Clean) and perturbed accuracy (Pert.) of the classifiers and of the Naïve/FD INR surrogates on ShapeNet-10.

| Dataset | Attack | Inr2Vec | | DWS | | NFN | | NFT | | ScaleGMN | |
|---|---|---|---|---|---|---|---|---|---|---|---|
| | | Clean | Pert. | Clean | Pert. | Clean | Pert. | Clean | Pert. | Clean | Pert. |
| ShapeNet-10 | *Classifier* | 92.7% | | 76.4% | | 87.4% | | 92.9% | | 88.8% | |
| | Naïve INR | 80.6% | **53.9%** | 54.3% | 44.6% | 80.9% | **52.2%** | 73.4% | **37.4%** | 79.8% | **50.5%** |
| | FD INR | 78.5% | 55.9% | 51.3% | **42.9%** | 76.9% | 54.0% | 75.5% | 51.4% | 78.9% | 60.7% |

We report the surrogate clean and perturbed accuracy on ShapeNet-10 in Table 10, analogous to Table 9 for the 2D benchmarks. Surrogate-based attacks substantially reduce accuracy across all five classifiers, demonstrating that the framework's effectiveness extends from 2D images to 3D point clouds.

## C.2. Effect of Finite-Difference Supervision

To isolate how gradient-approximation quality affects attack strength, we ablate $\lambda_{\mathrm{der}}$—the weight of the finite-difference Jacobian supervision in the FD surrogate loss—while fixing the other loss weights, on MNIST. The third loss term directly supervises the surrogate's Jacobian, so a lower Jacobian loss indicates closer alignment with the true INR fitting gradient.

*Table 11.* **Effect of finite-difference supervision strength on attack effectiveness on MNIST.** We vary $\lambda_{\mathrm{der}}$, the weight of the finite-difference Jacobian supervision, while fixing the other loss weights. Lower Jacobian loss indicates closer alignment with the true INR fitting gradient. Stronger supervision consistently lowers both the Jacobian loss and the perturbed accuracy (Adv. Acc), while surrogate clean accuracy stays stable.

| Classifier | $\lambda_{\mathrm{der}}$ | Surr. Clean | Jac. Loss | Adv. Acc |
|---|---|---|---|---|
| **Inr2Vec** | 0.01 | 97.05% | 0.403 | 0.02% |
|  | 0.1 | 97.38% | 0.350 | 0.03% |
|  | 1.0 | 97.85% | 0.337 | 0.00% |
|  | 10.0 | 97.88% | 0.333 | 0.00% |
| **DWS** | 0.01 | 96.77% | 0.432 | 4.44% |
|  | 0.1 | 97.63% | 0.360 | 7.50% |
|  | 1.0 | 97.09% | 0.337 | 0.11% |
|  | 10.0 | 97.83% | 0.334 | 0.00% |
| **NFN** | 0.01 | 95.78% | 0.408 | 0.03% |
|  | 0.1 | 94.82% | 0.415 | 0.02% |
|  | 1.0 | 96.09% | 0.366 | 0.01% |
|  | 10.0 | 94.54% | 0.336 | 0.00% |
| **NFT** | 0.01 | 97.85% | 0.357 | 16.62% |
|  | 0.1 | 97.46% | 0.345 | 0.98% |
|  | 1.0 | 97.86% | 0.336 | 0.17% |
|  | 10.0 | 97.97% | 0.334 | 0.12% |
| **ScaleGMN** | 0.01 | 96.78% | 0.417 | 22.83% |
|  | 0.1 | 97.89% | 0.348 | 19.14% |
|  | 1.0 | 97.65% | 0.335 | 12.75% |
|  | 10.0 | 98.22% | 0.333 | 0.07% |

Table 11 shows that better Jacobian alignment consistently yields stronger attacks (lower Adv. Acc), most clearly for classifiers not already near zero, i.e., NFT, ScaleGMN, and DWS, while surrogate clean accuracy stays stable. This indicates that the near-zero accuracy of our best surrogates reflects faithful gradient approximation to the real pipeline rather than a surrogate-specific artifact.

## C.3. Impact of INR Configuration

We evaluate how INR architecture affects adversarial robustness by varying depth, width, and frequency scale. Tables 12 and 13 report surrogate accuracy and perturbed accuracy across different numbers of hidden layers. Tables 14 and 15 show results for varying hidden units per layer. Tables 16 and 17 present results under different $\omega_0$ values.

*Table 12.* **Surrogate clean accuracy (%) for depth ablation.** "–" indicates not applicable.

| Dataset | Attack | Inr2Vec 0 | Inr2Vec 1 | Inr2Vec 3 | DWS 0 | DWS 1 | DWS 3 | NFN 0 | NFN 1 | NFN 3 | NFT 0 | NFT 1 | NFT 3 | ScaleGMN 0 | ScaleGMN 1 | ScaleGMN 3 |
|---|---|---|---|---|---|---|---|---|---|---|---|---|---|---|---|---|
| MNIST | Hypernetwork | – | 82.62% | 72.19% | – | 29.84% | 21.43% | 91.29% | 81.11% | 79.82% | 94.70% | 98.89% | 97.40% | 88.48% | 85.19% | 11.51% |
| | Naïve INR | – | 95.57% | 97.21% | – | 97.17% | 93.34% | 95.14% | 95.27% | 97.24% | 93.50% | 97.65% | 97.80% | 94.42% | 95.92% | 94.42% |
| | FD INR | – | 96.01% | 94.55% | – | 96.36% | 94.87% | 95.15% | 97.85% | 97.51% | 92.04% | 97.51% | 97.70% | 41.07% | 96.15% | 96.24% |
| | Naïve Emb. | – | 97.53% | 96.88% | – | – | – | – | – | – | 98.33% | 98.49% | 98.22% | – | – | – |
| | FD Emb. | – | 95.98% | 96.12% | – | – | – | – | – | – | 94.22% | 97.13% | 97.60% | – | – | – |
| F-MNIST | Hypernetwork | – | 82.24% | 58.04% | – | 25.74% | 27.41% | 85.86% | 86.31% | 69.81% | 87.30% | 91.69% | 83.68% | 80.04% | 52.54% | 29.80% |
| | Naïve INR | – | 87.84% | 88.32% | – | 88.73% | 83.43% | 82.85% | 88.41% | 84.95% | 84.40% | 88.37% | 88.30% | 83.56% | 88.59% | 83.56% |
| | FD INR | – | 87.21% | 86.77% | – | 88.61% | 64.06% | 80.96% | 86.26% | 85.06% | 77.80% | 88.82% | 88.10% | 57.55% | 82.23% | 86.83% |
| | Naïve Emb. | – | 88.66% | 86.23% | – | – | – | – | – | – | 88.48% | 84.51% | 87.28% | – | – | – |
| | FD Emb. | – | 84.93% | 87.44% | – | – | – | – | – | – | 73.05% | 86.51% | 87.34% | – | – | – |
| CIFAR-10 | Hypernetwork | – | 45.21% | 37.37% | – | 20.42% | 19.80% | 46.00% | 51.18% | 30.82% | 51.12% | 68.67% | 65.75% | 53.28% | 41.87% | 14.69% |
| | Naïve INR | – | 49.51% | 50.00% | – | 50.32% | 48.12% | 44.80% | 51.72% | 50.49% | 39.60% | 52.07% | 52.10% | 42.89% | 51.62% | 48.78% |
| | FD INR | – | 51.02% | 52.17% | – | 49.37% | 52.22% | 38.09% | 51.96% | 51.86% | 34.10% | 52.60% | 53.50% | 39.05% | 49.78% | 49.46% |
| | Naïve Emb. | – | 51.58% | 52.43% | – | – | – | – | – | – | 47.58% | 47.44% | 54.29% | – | – | – |
| | FD Emb. | – | 43.83% | 54.49% | – | – | – | – | – | – | 40.92% | 47.63% | 53.16% | – | – | – |

*Table 13.* **Impact of INR depth.** Perturbed accuracy (%) under each depth configuration. "–" indicates not applicable.

| Dataset | Attack | Inr2Vec 0 | Inr2Vec 1 | Inr2Vec 3 | DWS 0 | DWS 1 | DWS 3 | NFN 0 | NFN 1 | NFN 3 | NFT 0 | NFT 1 | NFT 3 | ScaleGMN 0 | ScaleGMN 1 | ScaleGMN 3 |
|---|---|---|---|---|---|---|---|---|---|---|---|---|---|---|---|---|
| MNIST | Hypernetwork | – | 0.01% | 0.03% | – | 0.03% | 0.25% | 1.78% | 0.17% | 0.01% | 0.00% | 0.00% | 0.00% | 0.00% | 0.09% | 1.00% |
| | Naïve INR | – | 2.19% | 0.03% | – | 10.12% | 3.95% | 0.16% | 0.29% | 0.56% | 0.40% | 20.57% | 9.80% | 0.00% | 7.16% | 0.00% |
| | FD INR | – | 1.91% | 2.55% | – | 0.01% | 2.95% | 0.00% | 0.40% | 0.00% | 0.02% | 0.00% | 0.00% | 3.75% | 0.00% | 24.02% |
| | Naïve Emb. | – | 0.69% | 0.65% | – | – | – | – | – | – | 0.00% | 0.00% | 0.02% | – | – | – |
| | FD Emb. | – | 2.08% | 0.09% | – | – | – | – | – | – | 0.00% | 0.00% | 0.00% | – | – | – |
| F-MNIST | Hypernetwork | – | 0.13% | 14.95% | – | 15.51% | 15.97% | 18.67% | 0.76% | 29.03% | 31.22% | 0.78% | 1.00% | 51.05% | 30.44% | 21.24% |
| | Naïve INR | – | 17.91% | 1.21% | – | 20.82% | 0.31% | 17.78% | 5.68% | 32.61% | 9.50% | 12.41% | 2.15% | 25.95% | 13.01% | 25.95% |
| | FD INR | – | 13.89% | 11.03% | – | 15.15% | 0.04% | 61.84% | 0.27% | 27.19% | 61.32% | 3.83% | 42.20% | 46.44% | 10.44% | 55.25% |
| | Naïve Emb. | – | 21.63% | 6.13% | – | – | – | – | – | – | 4.68% | 9.40% | 2.31% | – | – | – |
| | FD Emb. | – | 27.20% | 11.93% | – | – | – | – | – | – | 18.06% | 21.53% | 10.91% | – | – | – |
| CIFAR-10 | Hypernetwork | – | 0.48% | 0.76% | – | 0.38% | 0.31% | 0.05% | 0.40% | 0.39% | 0.04% | 0.15% | 0.08% | 1.17% | 1.07% | 7.92% |
| | Naïve INR | – | 0.46% | 0.76% | – | 3.70% | 4.60% | 0.07% | 4.95% | 0.03% | 0.00% | 4.13% | 12.00% | 0.01% | 3.96% | 0.43% |
| | FD INR | – | 5.22% | 2.33% | – | 2.84% | 1.63% | 11.87% | 0.15% | 0.02% | 3.70% | 0.19% | 0.40% | 9.21% | 0.25% | 0.09% |
| | Naïve Emb. | – | 4.58% | 1.89% | – | – | – | – | – | – | 0.01% | 0.82% | 0.07% | – | – | – |
| | FD Emb. | – | 13.59% | 3.55% | – | – | – | – | – | – | 0.81% | 2.13% | 0.32% | – | – | – |

*Table 14.* **Surrogate clean accuracy (%) for width ablation.** "–" indicates not applicable.

| Dataset | Attack | Inr2Vec 16 | Inr2Vec 32 | Inr2Vec 64 | DWS 16 | DWS 32 | DWS 64 | NFN 16 | NFN 32 | NFN 64 | NFT 16 | NFT 32 | NFT 64 | ScaleGMN 16 | ScaleGMN 32 | ScaleGMN 64 |
|---|---|---|---|---|---|---|---|---|---|---|---|---|---|---|---|---|
| MNIST | Hypernetwork | 67.36% | 82.62% | 86.84% | 34.78% | 29.84% | 32.52% | 93.87% | 81.11% | 93.33% | 97.52% | 98.89% | 98.29% | 55.70% | 85.19% | 89.13% |
| | Naïve INR | 97.00% | 95.57% | 97.20% | 96.95% | 97.17% | 85.71% | 96.93% | 95.27% | 95.21% | 97.00% | 97.65% | 21.00% | 71.91% | 95.92% | 96.90% |
| | FD INR | 79.20% | 96.01% | 84.30% | 83.09% | 96.36% | 84.28% | 96.15% | 97.85% | 95.63% | 96.50% | 97.51% | 25.20% | 40.02% | 96.15% | 93.68% |
| | Naïve Emb. | 98.35% | 97.53% | 97.49% | – | – | – | – | – | – | 94.23% | 98.49% | 20.48% | – | – | – |
| | FD Emb. | 95.74% | 95.98% | 96.76% | – | – | – | – | – | – | 83.49% | 97.13% | 96.84% | – | – | – |
| F-MNIST | Hypernetwork | 61.74% | 82.24% | 73.69% | 10.17% | 25.74% | 24.86% | 83.59% | 86.31% | 87.50% | 87.77% | 91.69% | 88.11% | 40.88% | 52.54% | 18.84% |
| | Naïve INR | 87.40% | 87.84% | 88.40% | 86.15% | 88.73% | 71.08% | 82.72% | 88.41% | 87.28% | 70.10% | 88.37% | 88.70% | 86.78% | 88.59% | 86.76% |
| | FD INR | 82.40% | 87.21% | 87.30% | 72.59% | 88.61% | 60.60% | 87.69% | 86.26% | 87.52% | 79.60% | 88.82% | 88.40% | 76.03% | 82.23% | 10.00% |
| | Naïve Emb. | 87.82% | 88.66% | 87.87% | – | – | – | – | – | – | 84.51% | 87.69% | 87.83% | – | – | – |
| | FD Emb. | 84.08% | 84.93% | 86.04% | – | – | – | – | – | – | 86.51% | 87.78% | 87.76% | – | – | – |
| CIFAR-10 | Hypernetwork | 35.06% | 45.21% | 41.28% | 33.68% | 20.42% | 21.99% | 41.81% | 51.18% | 50.81% | 33.89% | 68.67% | 37.59% | 51.69% | 41.87% | 32.44% |
| | Naïve INR | 34.60% | 49.51% | 51.10% | 51.84% | 50.32% | 10.00% | 47.56% | 51.72% | 51.30% | 46.30% | 52.07% | 47.20% | 50.07% | 51.62% | 50.97% |
| | FD INR | 35.50% | 51.02% | 52.70% | 36.67% | 49.37% | 34.03% | 45.51% | 51.96% | 51.09% | 37.00% | 52.60% | 50.30% | 39.86% | 49.78% | 51.84% |
| | Naïve Emb. | 50.29% | 51.58% | 51.57% | – | – | – | – | – | – | 51.60% | 47.44% | 46.26% | – | – | – |
| | FD Emb. | 42.56% | 43.83% | 53.26% | – | – | – | – | – | – | 34.08% | 47.63% | 44.87% | – | – | – |

*Table 15.* **Impact of INR width.** Perturbed accuracy (%) under each width configuration. "–" indicates not applicable.

| Dataset | Attack | Inr2Vec | | | DWS | | | NFN | | | NFT | | | ScaleGMN | | |
|---|---|---|---|---|---|---|---|---|---|---|---|---|---|---|---|---|
| | | 16 | 32 | 64 | 16 | 32 | 64 | 16 | 32 | 64 | 16 | 32 | 64 | 16 | 32 | 64 |
| **MNIST** | Hypernetwork | 0.07% | 0.01% | 0.00% | 0.22% | 0.03% | 0.21% | 5.87% | 0.17% | 0.00% | 0.00% | 0.00% | 0.00% | 0.29% | 0.09% | 0.03% |
| | Naïve INR | 2.46% | 2.19% | 0.23% | 0.07% | 10.12% | 0.66% | 7.07% | 0.29% | 1.80% | 2.34% | 20.57% | 17.60% | 0.06% | 7.16% | 10.12% |
| | FD INR | 0.62% | 1.91% | 0.00% | 1.11% | 0.01% | 0.00% | 0.03% | 0.40% | 0.00% | 0.00% | 0.00% | 3.40% | 0.77% | 0.00% | 0.00% |
| | Naïve Emb. | 0.20% | 0.69% | 0.99% | – | – | – | – | – | – | 0.22% | 0.00% | 0.00% | – | – | – |
| | FD Emb. | 7.09% | 2.08% | 0.36% | – | – | – | – | – | – | 0.02% | 0.00% | 0.00% | – | – | – |
| **F-MNIST** | Hypernetwork | 25.02% | 0.13% | 19.66% | 2.00% | 15.51% | 13.23% | 35.93% | 0.76% | 15.81% | 34.80% | 0.78% | 24.54% | 28.57% | 30.44% | 17.01% |
| | Naïve INR | 2.88% | 17.91% | 1.61% | 4.73% | 20.82% | 16.27% | 43.15% | 5.68% | 37.97% | 36.60% | 12.41% | 5.50% | 50.22% | 13.01% | 60.41% |
| | FD INR | 33.79% | 13.89% | 16.44% | 32.92% | 15.15% | 19.61% | 63.47% | 0.27% | 33.79% | 63.75% | 3.83% | 44.44% | 58.69% | 10.44% | 10.00% |
| | Naïve Emb. | 1.34% | 21.63% | 2.19% | – | – | – | – | – | – | 9.40% | 45.23% | 7.90% | – | – | – |
| | FD Emb. | 36.06% | 27.20% | 14.29% | – | – | – | – | – | – | 21.53% | 48.65% | 7.85% | – | – | – |
| **CIFAR-10** | Hypernetwork | 1.64% | 0.48% | 0.76% | 0.65% | 0.38% | 0.33% | 1.88% | 0.40% | 0.07% | 0.68% | 0.15% | 0.06% | 0.39% | 1.07% | 4.76% |
| | Naïve INR | 0.17% | 0.46% | 0.48% | 1.93% | 3.70% | 10.00% | 1.54% | 4.95% | 0.06% | 0.02% | 4.13% | 14.10% | 2.78% | 3.96% | 0.16% |
| | FD INR | 10.21% | 5.22% | 3.06% | 9.97% | 2.84% | 0.24% | 4.35% | 0.15% | 0.01% | 16.08% | 0.19% | 0.10% | 1.61% | 0.25% | 0.17% |
| | Naïve Emb. | 1.04% | 4.58% | 0.64% | – | – | – | – | – | – | 0.18% | 0.82% | 0.09% | – | – | – |
| | FD Emb. | 16.36% | 13.59% | 0.50% | – | – | – | – | – | – | 13.85% | 2.13% | 0.04% | – | – | – |

*Table 16.* **Surrogate clean accuracy (%) for frequency scale ($\omega_0$) ablation.** "–" indicates not applicable.

| Dataset | Attack | Inr2Vec | | | DWS | | | NFN | | | NFT | | | ScaleGMN | | |
|---|---|---|---|---|---|---|---|---|---|---|---|---|---|---|---|---|
| | | 10 | 30 | 60 | 10 | 30 | 60 | 10 | 30 | 60 | 10 | 30 | 60 | 10 | 30 | 60 |
| **MNIST** | Hypernetwork | 42.87% | 82.62% | 89.14% | 19.54% | 29.84% | 17.98% | 82.78% | 81.11% | 96.66% | 98.86% | 98.89% | 98.22% | 82.71% | 85.19% | 7.55% |
| | Naïve INR | 97.47% | 95.57% | 96.48% | 93.07% | 97.17% | 94.90% | 94.40% | 95.27% | 93.35% | 97.20% | 97.65% | 97.00% | 92.30% | 95.92% | 91.81% |
| | FD INR | 95.70% | 96.01% | 84.40% | 75.53% | 96.36% | 3.66% | 96.56% | 97.85% | 81.17% | 97.20% | 97.51% | 87.70% | 86.00% | 96.15% | 88.33% |
| | Naïve Emb. | 97.80% | 97.53% | 96.70% | – | – | – | – | – | – | 88.45% | 98.49% | 98.44% | – | – | – |
| | FD Emb. | 98.03% | 95.98% | 56.78% | – | – | – | – | – | – | 97.28% | 97.13% | 96.90% | – | – | – |
| **F-MNIST** | Hypernetwork | 41.71% | 82.24% | 69.84% | 41.00% | 25.74% | 22.58% | 77.43% | 86.31% | 88.06% | 79.83% | 91.69% | – | 44.97% | 52.54% | 51.53% |
| | Naïve INR | 87.02% | 87.84% | 87.64% | 84.82% | 88.73% | 75.62% | 73.26% | 88.41% | 82.05% | 88.80% | 88.37% | 88.20% | 85.40% | 88.59% | 84.02% |
| | FD INR | 81.20% | 87.21% | 75.70% | 78.83% | 88.61% | 24.05% | 85.53% | 86.26% | 44.61% | 88.30% | 88.82% | 82.60% | 72.90% | 82.23% | 55.73% |
| | Naïve Emb. | 87.90% | 88.66% | 87.20% | – | – | – | – | – | – | 88.45% | 84.51% | 88.02% | – | – | – |
| | FD Emb. | 87.15% | 84.93% | 85.47% | – | – | – | – | – | – | 84.03% | 86.51% | 87.05% | – | – | – |
| **CIFAR-10** | Hypernetwork | 24.56% | 45.21% | 42.56% | 17.99% | 20.42% | 18.63% | 44.31% | 51.18% | 49.22% | 59.80% | 68.67% | 65.89% | 32.88% | 41.87% | 41.07% |
| | Naïve INR | 50.98% | 49.51% | 49.22% | 48.47% | 50.32% | 48.33% | 47.02% | 51.72% | 45.63% | 49.10% | 52.07% | 51.80% | 41.90% | 51.62% | 46.46% |
| | FD INR | 44.70% | 51.02% | 41.95% | 49.74% | 49.37% | 45.13% | 48.33% | 51.96% | 48.22% | 39.40% | 52.60% | 45.30% | 40.20% | 49.78% | 40.90% |
| | Naïve Emb. | 50.00% | 51.58% | 50.40% | – | – | – | – | – | – | 45.93% | 47.44% | 52.85% | – | – | – |
| | FD Emb. | 53.70% | 43.83% | 47.53% | – | – | – | – | – | – | 43.52% | 47.63% | 48.89% | – | – | – |

*Table 17.* **Impact of frequency scale ($\omega_0$).** Perturbed accuracy (%) under each $\omega_0$ configuration. "–" indicates not applicable.

| Dataset | Attack | Inr2Vec | | | DWS | | | NFN | | | NFT | | | ScaleGMN | | |
|---|---|---|---|---|---|---|---|---|---|---|---|---|---|---|---|---|
| | | 10 | 30 | 60 | 10 | 30 | 60 | 10 | 30 | 60 | 10 | 30 | 60 | 10 | 30 | 60 |
| **MNIST** | Hypernetwork | 0.04% | 0.01% | 0.00% | 0.05% | 0.03% | 0.19% | 1.33% | 0.17% | 0.00% | 0.00% | 0.00% | 0.00% | 0.03% | 0.09% | 0.08% |
| | Naïve INR | 0.01% | 2.19% | 1.12% | 0.17% | 10.12% | 0.78% | 8.46% | 0.29% | 0.02% | 21.30% | 20.57% | 13.10% | 0.00% | 7.16% | 0.00% |
| | FD INR | 0.53% | 1.91% | 0.11% | 0.46% | 0.01% | 0.26% | 0.01% | 0.40% | 0.00% | 0.01% | 0.00% | 0.00% | 0.65% | 0.00% | 0.01% |
| | Naïve Emb. | 0.15% | 0.69% | 1.80% | – | – | – | – | – | – | 53.11% | 0.00% | 0.00% | – | – | – |
| | FD Emb. | 0.52% | 2.08% | 15.59% | – | – | – | – | – | – | 0.13% | 0.00% | 0.01% | – | – | – |
| **F-MNIST** | Hypernetwork | 7.29% | 0.13% | 14.86% | 1.29% | 15.51% | 1.18% | 38.03% | 0.76% | 22.93% | 1.03% | 0.78% | – | 25.04% | 30.44% | 31.14% |
| | Naïve INR | 6.40% | 17.91% | 5.18% | 2.14% | 20.82% | 0.29% | 15.60% | 5.68% | 8.94% | 12.60% | 12.41% | 2.05% | 1.71% | 13.01% | 0.18% |
| | FD INR | 31.29% | 13.89% | 13.53% | 25.41% | 15.15% | 5.67% | 14.94% | 0.27% | 22.75% | 20.81% | 3.83% | 8.78% | 22.22% | 10.44% | 9.35% |
| | Naïve Emb. | 2.91% | 21.63% | 3.94% | – | – | – | – | – | – | 53.11% | 9.40% | 52.06% | – | – | – |
| | FD Emb. | 19.11% | 27.20% | 19.84% | – | – | – | – | – | – | 11.91% | 21.53% | 20.10% | – | – | – |
| **CIFAR-10** | Hypernetwork | 1.17% | 0.48% | 1.38% | 1.31% | 0.38% | 0.53% | 0.47% | 0.40% | 0.41% | 0.00% | 0.15% | 0.42% | 3.19% | 1.07% | 2.99% |
| | Naïve INR | 0.30% | 0.46% | 0.50% | 0.00% | 3.70% | 0.01% | 0.00% | 4.95% | 0.01% | 3.06% | 4.13% | 7.30% | 0.02% | 3.96% | 0.01% |
| | FD INR | 6.40% | 5.22% | 5.58% | 3.37% | 2.84% | 5.19% | 4.32% | 0.15% | 2.72% | 5.99% | 0.19% | 3.85% | 6.37% | 0.25% | 3.70% |
| | Naïve Emb. | 2.07% | 4.58% | 0.09% | – | – | – | – | – | – | 0.21% | 0.82% | 0.03% | – | – | – |
| | FD Emb. | 5.89% | 13.59% | 8.96% | – | – | – | – | – | – | 1.31% | 2.13% | 0.17% | – | – | – |

## C.4. Impact of the Embedding dimensionality

Tables 18 and 19 show how embedding dimension affects robustness for encoder-based pipelines, i.e., Inr2Vec and NFT.

*Table 18.* **Surrogate clean accuracy (%) for embedding dimension ablation.** "–" indicates not applicable.

| Dataset | Attack | Inr2Vec | | | | NFT | | | |
|---|---|---|---|---|---|---|---|---|---|
| | | 64 | 128 | 256 | 512 | 64 | 128 | 256 | 512 |
| **MNIST** | *Classifier* | 81.49% | 89.40% | 88.60% | 89.40% | 97.67% | 97.66% | 97.64% | 97.09% |
| | Hypernetwork | 82.26% | 82.26% | 82.62% | 82.26% | 98.37% | 98.73% | 98.89% | 98.59% |
| | Naïve INR | 94.52% | 93.51% | 95.57% | 93.49% | 97.15% | 96.15% | 97.65% | 96.10% |
| | FD INR | 92.64% | 92.59% | 96.01% | 91.48% | 97.57% | 97.64% | 97.51% | 97.23% |
| | Naïve Emb. | 98.18% | 97.27% | 97.67% | 97.42% | 98.66% | 98.74% | 98.50% | 98.17% |
| | FD Emb. | 97.90% | 97.57% | 95.98% | 97.43% | 98.35% | 98.39% | 97.13% | 98.32% |
| **F-MNIST** | *Classifier* | 76.33% | 77.82% | 78.31% | 79.16% | 83.59% | 83.88% | 83.22% | 83.65% |
| | Hypernetwork | 67.85% | 67.85% | 82.24% | 67.85% | 90.42% | 90.42% | 91.69% | 90.42% |
| | Naïve INR | 86.05% | 83.20% | 87.84% | 81.41% | 86.60% | 86.60% | 88.37% | 86.60% |
| | FD INR | 83.82% | 83.35% | 87.21% | 82.27% | 86.93% | 86.93% | 88.82% | 86.93% |
| | Naïve Emb. | 88.32% | 88.61% | 88.66% | 87.67% | 87.73% | 87.73% | 84.38% | 87.73% |
| | FD Emb. | 87.50% | 88.37% | 88.44% | 87.29% | 89.17% | 89.17% | 86.48% | 89.17% |
| **CIFAR-10** | *Classifier* | 34.71% | 38.50% | 35.13% | 40.16% | 47.37% | 48.35% | 48.16% | 48.16% |
| | Hypernetwork | 38.59% | 35.38% | 45.21% | 37.08% | 65.75% | 64.82% | 68.67% | 53.46% |
| | Naïve INR | 46.90% | 47.31% | 49.51% | 47.80% | 49.75% | 48.62% | 52.07% | 45.85% |
| | FD INR | 49.70% | 48.01% | 51.02% | 47.71% | 52.60% | 14.79% | 52.60% | 14.21% |
| | Naïve Emb. | 51.57% | 50.84% | 51.93% | 51.73% | 54.32% | 54.68% | 47.29% | 47.98% |
| | FD Emb. | 51.90% | 52.35% | 43.68% | 52.60% | 54.78% | 54.50% | 47.89% | 51.60% |

*Table 19.* **Impact of embedding dimension.** Perturbed accuracy (%) under each dimension configuration.

| Dataset | Attack | Inr2Vec | | | | NFT | | | |
|---|---|---|---|---|---|---|---|---|---|
| | | 64 | 128 | 256 | 512 | 64 | 128 | 256 | 512 |
| **MNIST** | *Classifier* | 81.49% | 89.40% | 88.60% | 89.40% | 97.67% | 97.66% | 97.64% | 97.09% |
| | Hypernetwork | 0.01% | 0.01% | 0.01% | 0.01% | 0.00% | 0.00% | 0.00% | 0.00% |
| | Naïve INR | 0.00% | 0.10% | 2.19% | 0.02% | 0.04% | 0.00% | 20.57% | 0.10% |
| | FD INR | 0.03% | 0.12% | 1.91% | 0.05% | 0.02% | 0.00% | 0.00% | 0.00% |
| | Naïve Emb. | 0.00% | 0.10% | 0.69% | 1.63% | 0.00% | 0.01% | 0.00% | 0.06% |
| | FD Emb. | 0.56% | 0.43% | 2.08% | 3.15% | 0.00% | 0.00% | 0.00% | 0.02% |
| **F-MNIST** | *Classifier* | 76.33% | 77.82% | 78.31% | 79.16% | 83.59% | 83.88% | 83.22% | 83.65% |
| | Hypernetwork | 23.24% | 23.24% | 0.13% | 23.24% | 0.23% | 0.23% | 0.78% | 0.23% |
| | Naïve INR | 13.36% | 9.29% | 17.91% | 11.05% | 11.37% | 11.37% | 12.41% | 11.37% |
| | FD INR | 22.09% | 22.15% | 13.89% | 21.39% | 16.19% | 16.19% | 3.83% | 16.19% |
| | Naïve Emb. | 2.52% | 2.37% | 21.63% | 3.67% | 10.63% | 10.63% | 9.40% | 10.63% |
| | FD Emb. | 8.36% | 5.60% | 13.17% | 4.02% | 15.67% | 15.67% | 21.53% | 15.67% |
| **CIFAR-10** | *Classifier* | 34.71% | 38.50% | 35.13% | 40.16% | 47.37% | 48.35% | 48.16% | 48.16% |
| | Hypernetwork | 0.68% | 0.57% | 0.48% | 0.63% | 0.04% | 0.18% | 0.15% | 0.63% |
| | Naïve INR | 2.51% | 1.90% | 0.46% | 1.97% | 3.34% | 4.62% | 4.13% | 7.80% |
| | FD INR | 4.62% | 6.00% | 5.22% | 7.91% | 3.96% | 4.03% | 0.19% | 7.08% |
| | Naïve Emb. | 0.67% | 0.92% | 4.58% | 0.92% | 0.88% | 0.83% | 0.82% | 4.63% |
| | FD Emb. | 3.16% | 2.78% | 13.61% | 0.85% | 0.86% | 1.30% | 2.13% | 5.09% |

## C.5. Impact of the Number of INRs

Tables 20 and 21 report results when fitting multiple INRs per image with different random initializations.

*Table 20.* **Surrogate clean accuracy (%) for the number of INRs ablation.** "–" indicates not applicable.

| Dataset | Attack | Inr2Vec | | | DWS | | | NFN | | | NFT | | | ScaleGMN | | |
|---|---|---|---|---|---|---|---|---|---|---|---|---|---|---|---|---|
| | | 1 | 3 | 5 | 1 | 3 | 5 | 1 | 3 | 5 | 1 | 3 | 5 | 1 | 3 | 5 |
| MNIST | *Classifier* | 88.99% | 82.91% | 80.98% | 75.48% | 76.95% | 76.33% | 90.76% | 88.32% | 88.21% | 98.36% | 98.17% | 98.05% | 94.75% | 94.16% | 95.03% |
| | Hypernetwork | 82.62% | 71.68% | 70.32% | 29.84% | 28.69% | 39.69% | 81.11% | 93.24% | 94.39% | 98.89% | 97.79% | 98.19% | 85.19% | 54.87% | 75.99% |
| | Naïve INR | 95.57% | 90.77% | 90.60% | 97.17% | 55.50% | 71.40% | 95.27% | 96.09% | 95.36% | 97.65% | 97.33% | 96.79% | 95.92% | 96.84% | 95.14% |
| | FD INR | 96.01% | 93.29% | 88.04% | 96.36% | 67.16% | 60.50% | 97.85% | 96.78% | 96.24% | 97.51% | 96.85% | 96.51% | 96.15% | 96.45% | 97.09% |
| | Naïve Emb. | 97.53% | 98.07% | 98.07% | – | – | – | – | – | – | 98.49% | 97.65% | 97.61% | – | – | – |
| | FD Emb. | 95.98% | 91.56% | 91.56% | – | – | – | – | – | – | 97.13% | 70.84% | 68.54% | – | – | – |
| F-MNIST | *Classifier* | 78.67% | 77.30% | 75.82% | 74.63% | 74.50% | 74.63% | 79.27% | 78.71% | 79.61% | 83.79% | 83.39% | 84.07% | 82.78% | 82.03% | 81.93% |
| | Hypernetwork | 82.24% | 44.62% | 49.75% | 25.74% | 18.84% | 29.70% | 86.31% | 66.15% | 85.95% | 91.69% | 97.79% | 98.19% | 52.54% | 66.09% | 61.41% |
| | Naïve INR | 87.84% | 84.91% | 85.51% | 88.73% | 52.09% | 52.42% | 88.41% | 94.60% | 86.40% | 88.37% | 88.93% | 87.46% | 88.59% | 84.54% | 84.74% |
| | FD INR | 87.21% | 78.93% | 72.10% | 88.61% | 39.28% | 27.30% | 86.26% | 91.36% | 85.48% | 88.82% | 88.63% | 88.28% | 82.23% | 80.30% | 82.94% |
| | Naïve Emb. | 88.66% | 84.91% | 88.44% | – | – | – | – | – | – | 84.51% | 88.81% | 88.77% | – | – | – |
| | FD Emb. | 84.93% | 81.29% | 87.01% | – | – | – | – | – | – | 86.51% | 86.02% | 85.59% | – | – | – |
| CIFAR-10 | *Classifier* | 35.13% | 41.21% | 32.54% | 39.63% | 31.63% | 41.06% | 41.50% | 44.95% | 45.69% | 48.87% | 56.10% | 58.13% | 48.56% | 51.34% | 52.12% |
| | Hypernetwork | 45.21% | 46.38% | 53.56% | 20.42% | 51.14% | 33.87% | 51.18% | 55.81% | 58.70% | 68.67% | 59.56% | 54.80% | 41.87% | 41.88% | 45.04% |
| | Naïve INR | 49.51% | 42.78% | 41.21% | 50.32% | 50.35% | 42.55% | 51.72% | 49.70% | 47.53% | 52.07% | 43.14% | 38.14% | 51.62% | 49.69% | 48.98% |
| | FD INR | 51.02% | 41.29% | 42.00% | 49.37% | 51.79% | 47.89% | 51.96% | 51.06% | 54.10% | 52.60% | 43.64% | 46.30% | 49.78% | 51.04% | 48.92% |
| | Naïve Emb. | 51.58% | 45.10% | 50.11% | – | – | – | – | – | – | 47.44% | 52.29% | 53.49% | – | – | – |
| | FD Emb. | 43.83% | 44.91% | 51.75% | – | – | – | – | – | – | 47.63% | 55.54% | 30.46% | – | – | – |

*Table 21.* **Full results on the number of INRs.** We report perturbed accuracy (%) for different numbers of INRs fitted per image. "–" indicates the attack is not applicable.

| Dataset | Attack | Inr2Vec | | | DWS | | | NFN | | | NFT | | | ScaleGMN | | |
|---|---|---|---|---|---|---|---|---|---|---|---|---|---|---|---|---|
| | | 1 | 3 | 5 | 1 | 3 | 5 | 1 | 3 | 5 | 1 | 3 | 5 | 1 | 3 | 5 |
| MNIST | *Classifier* | 88.99% | 82.91% | 80.98% | 75.48% | 76.95% | 76.33% | 90.76% | 88.32% | 88.21% | 98.36% | 98.17% | 98.05% | 94.75% | 94.16% | 95.03% |
| | Hypernetwork | 0.01% | 0.14% | 0.02% | 0.03% | 0.63% | 0.11% | 0.17% | 0.59% | 0.12% | 0.00% | 0.00% | 0.00% | 0.09% | 0.06% | 0.06% |
| | Naïve INR | 2.19% | 5.95% | 9.70% | 10.12% | 7.11% | 3.82% | 0.29% | 0.18% | 1.64% | 20.57% | 8.41% | 11.46% | 7.16% | 15.36% | 8.58% |
| | FD INR | 1.91% | 13.20% | 8.64% | 0.01% | 6.49% | 7.98% | 0.40% | 0.79% | 3.24% | 0.00% | 2.31% | 4.59% | 0.00% | 2.68% | 0.22% |
| | Naïve Emb. | 0.69% | 1.25% | 1.25% | – | – | – | – | – | – | 0.00% | 0.06% | 0.09% | – | – | – |
| | FD Emb. | 2.08% | 1.01% | 1.01% | – | – | – | – | – | – | 0.00% | 1.63% | 4.04% | – | – | – |
| F-MNIST | *Classifier* | 78.67% | 77.30% | 75.82% | 74.63% | 74.50% | 74.63% | 79.27% | 78.71% | 79.61% | 83.79% | 83.39% | 84.07% | 82.78% | 82.03% | 81.93% |
| | Hypernetwork | 0.13% | 0.18% | 0.41% | 15.51% | 1.14% | 0.47% | 0.76% | 9.79% | 5.05% | 0.78% | 0.00% | 0.00% | 30.44% | 2.79% | 4.15% |
| | Naïve INR | 17.91% | 10.77% | 12.59% | 20.82% | 2.10% | 2.84% | 5.68% | 12.40% | 2.57% | 12.41% | 12.57% | 5.81% | 13.01% | 7.51% | 8.73% |
| | FD INR | 13.89% | 2.16% | 1.59% | 15.15% | 0.47% | 1.67% | 0.27% | 0.44% | 0.36% | 3.83% | 6.30% | 5.28% | 10.44% | 7.26% | 4.85% |
| | Naïve Emb. | 21.63% | 10.77% | 18.00% | – | – | – | – | – | – | 9.40% | 1.76% | 3.84% | – | – | – |
| | FD Emb. | 27.20% | 3.70% | 19.99% | – | – | – | – | – | – | 21.53% | 23.85% | 22.45% | – | – | – |
| CIFAR-10 | *Classifier* | 35.13% | 41.21% | 32.54% | 39.63% | 31.63% | 41.06% | 41.50% | 44.95% | 45.69% | 48.87% | 56.10% | 58.13% | 48.56% | 51.34% | 52.12% |
| | Hypernetwork | 0.48% | 0.60% | 0.34% | 0.38% | 0.34% | 0.56% | 0.40% | 0.00% | 0.00% | 0.15% | 0.00% | 0.00% | 1.07% | 0.63% | 0.12% |
| | Naïve INR | 0.46% | 3.06% | 5.28% | 3.70% | 1.43% | 1.66% | 4.95% | 1.36% | 4.43% | 4.13% | 5.67% | 6.45% | 3.96% | 1.21% | 1.74% |
| | FD INR | 5.22% | 5.90% | 8.83% | 2.84% | 1.73% | 2.23% | 0.15% | 1.13% | 4.43% | 0.19% | 4.01% | 3.65% | 0.25% | 1.32% | 1.91% |
| | Naïve Emb. | 4.58% | 4.01% | 3.03% | – | – | – | – | – | – | 0.82% | 0.21% | 0.31% | – | – | – |
| | FD Emb. | 13.59% | 5.02% | 2.96% | – | – | – | – | – | – | 2.13% | 4.77% | 8.61% | – | – | – |

## C.6. Impact on Attack Configuration

We evaluate how attack hyperparameters affect robustness under the FD INR surrogate attack. Figure 4 summarizes CIFAR-10 trends across attack configurations; Tables 22, 23, 24, 25, and 26 report full results across classifiers and datasets.

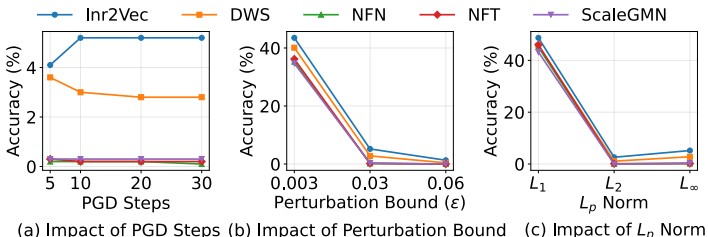

(a) Impact of PGD Steps (b) Impact of Perturbation Bound (c) Impact of $L_p$ Norm

*Figure 4.* **Impact of attack configurations on adversarial robustness.** We vary (a) the number of PGD steps, (b) perturbation bound $\varepsilon$, and (c) $L_p$-norm using FD INR surrogate attack on CIFAR-10.

We evaluate which hyperparameters most strongly influence attack strength and whether their effects align with prior findings on models trained with explicit representations. We examine the impact of attack iterations, choice of $\ell_p$ norms, and perturbation bounds. Figure 4 shows how attack hyperparameters affect robustness on CIFAR-10. Increasing PGD steps lead to stronger attack in general, however, Inr2Vec shows slight increase in perturbed accuracy at higher step counts. This may reflect their lower surrogate accuracy: when the surrogate gradient poorly approximates the true classifier's gradient, additional optimization steps do not guarantee stronger attacks and may even move samples away from effective adversarial directions. Larger perturbation bounds lead to stronger attacks as expected, with Acc dropping from 30–40% at $\varepsilon = 0.003$ to near zero at $\varepsilon = 0.06$. Among $\ell_p$ norms, $\ell_2$ and $\ell_\infty$ achieve similar effectiveness, while $\ell_1$ produces substantially weaker attacks, consistent with prior findings that $\ell_1$-PGD is less effective (Croce & Hein, 2021).

*Table 22.* **Impact of perturbation bound** ($\varepsilon$). We report perturbed accuracy (%) under each $\varepsilon$ value. "–" indicates the attack is not applicable.

| Dataset | Attack | Inr2Vec | | | DWS | | | NFN | | | NFT | | | ScaleGMN | | |
|---|---|---|---|---|---|---|---|---|---|---|---|---|---|---|---|---|
| | | .003 | .03 | .06 | .003 | .03 | .06 | .003 | .03 | .06 | .003 | .03 | .06 | .003 | .03 | .06 |
| **MNIST** | *Classifier* | | 89.4% | | | 75.5% | | | 88.5% | | | 97.6% | | | 94.5% | |
| | Hypernetwork | 80.1% | 49.0% | 16.7% | 34.9% | 22.4% | 12.8% | 80.1% | 63.3% | 34.5% | 98.7% | 95.4% | 79.2% | 83.7% | 70.3% | 46.9% |
| | Naïve INR | 95.3% | 89.8% | 76.7% | 96.7% | 91.1% | 77.5% | 94.4% | 80.5% | 44.8% | 97.5% | 92.6% | 79.3% | 95.3% | 88.1% | 73.3% |
| | FD INR | 95.9% | 90.2% | 77.9% | 95.8% | 84.4% | 55.1% | 97.3% | 82.8% | 43.3% | 97.0% | 84.9% | 47.0% | 95.4% | 80.2% | 42.4% |
| | Naïve Emb. | 97.4% | 93.7% | 85.2% | – | – | – | – | – | – | 98.2% | 94.2% | 80.7% | – | – | – |
| | FD Emb. | 95.4% | 91.6% | 83.2% | – | – | – | – | – | – | 96.8% | 90.7% | 72.6% | – | – | – |
| **F-MNIST** | *Classifier* | | 79.8% | | | 73.3% | | | 79.8% | | | 83.2% | | | 82.6% | |
| | Hypernetwork | 77.6% | 25.7% | 2.2% | 21.5% | 11.2% | 10.0% | 82.2% | 25.6% | 3.8% | 87.8% | 37.0% | 8.0% | 49.7% | 30.4% | 10.7% |
| | Naïve INR | 85.3% | 58.1% | 34.8% | 85.6% | 59.2% | 39.6% | 85.9% | 54.9% | 25.7% | 86.8% | 68.2% | 47.1% | 85.8% | 60.6% | 29.0% |
| | FD INR | 85.4% | 60.9% | 36.4% | 86.3% | 57.0% | 31.5% | 82.4% | 35.1% | 4.9% | 86.2% | 47.7% | 18.9% | 79.4% | 54.6% | 28.9% |
| | Naïve Emb. | 86.4% | 60.3% | 40.3% | – | – | – | – | – | – | 80.1% | 1.0% | 9.4% | – | – | – |
| | FD Emb. | 81.3% | 10.5% | 27.2% | – | – | – | – | – | – | 82.8% | 5.0% | 21.5% | – | – | – |
| **CIFAR-10** | *Classifier* | | 45.1% | | | 40.2% | | | 40.6% | | | 48.2% | | | 48.2% | |
| | Hypernetwork | 32.9% | 0.5% | 0.0% | 13.9% | 0.4% | 0.2% | 36.5% | 0.4% | 0.1% | 35.6% | 0.2% | 0.0% | – | 1.1% | 0.1% |
| | Naïve INR | 34.1% | 0.5% | 0.1% | 38.8% | 3.7% | 1.0% | 41.9% | 5.0% | 1.8% | 42.1% | 4.1% | 1.2% | 39.9% | 4.0% | 1.2% |
| | FD INR | 43.5% | 5.2% | 1.3% | 40.1% | 2.8% | 0.4% | 35.3% | 0.2% | 0.0% | 36.2% | 0.2% | 0.0% | 34.7% | 0.3% | 0.0% |
| | Naïve Emb. | 41.8% | 4.6% | 1.5% | – | – | – | – | – | – | 37.1% | 0.8% | 0.0% | – | – | – |
| | FD Emb. | 40.2% | 13.6% | 2.9% | – | – | – | – | – | – | 39.4% | 2.1% | 0.1% | – | – | – |

*Table 23.* **Surrogate clean accuracy (%) for $\ell_p$ norm ablation.** We report clean accuracy of each surrogate model. "–" indicates the surrogate is not applicable.

| Dataset | Surrogate | Inr2Vec | DWS | NFN | NFT | ScaleGMN |
|---|---|---|---|---|---|---|
| **MNIST** | Hypernetwork | 82.6% | 29.8% | 81.1% | 98.9% | 85.2% |
| | Naïve INR | 95.6% | 97.2% | 95.3% | 97.7% | 95.9% |
| | FD INR | 96.0% | 96.4% | 97.9% | 97.5% | 96.2% |
| | Naïve Emb. | 97.7% | – | – | 95.7% | – |
| | FD Emb. | 95.9% | – | – | 91.9% | – |
| **F-MNIST** | Hypernetwork | 82.2% | 25.7% | 86.3% | 86.5% | 52.5% |
| | Naïve INR | 87.8% | 88.7% | 88.4% | 98.5% | 88.6% |
| | FD INR | 87.2% | 88.6% | 86.3% | 97.1% | 82.2% |
| | Naïve Emb. | 88.7% | – | – | 84.4% | – |
| | FD Emb. | 84.7% | – | – | 86.5% | – |
| **CIFAR-10** | Hypernetwork | 45.2% | 20.4% | 51.2% | 68.7% | 41.9% |
| | Naïve INR | 49.5% | 50.3% | 51.7% | 52.1% | 51.6% |
| | FD INR | 51.0% | 49.4% | 52.0% | 52.6% | 49.8% |
| | Naïve Emb. | 51.9% | – | – | 47.3% | – |
| | FD Emb. | 43.7% | – | – | 47.9% | – |

*Table 24.* **Impact of $\ell_p$ norm.** We report perturbed accuracy (%) under each norm. Default is $\ell_\infty$. "–" indicates the attack is not applicable.

| Dataset | Attack | Inr2Vec $\ell_1$ | $\ell_2$ | $\ell_\infty$ | DWS $\ell_1$ | $\ell_2$ | $\ell_\infty$ | NFN $\ell_1$ | $\ell_2$ | $\ell_\infty$ | NFT $\ell_1$ | $\ell_2$ | $\ell_\infty$ | ScaleGMN $\ell_1$ | $\ell_2$ | $\ell_\infty$ |
|---|---|---|---|---|---|---|---|---|---|---|---|---|---|---|---|---|
| **MNIST** | *Classifier* | | 89.4% | | | 75.5% | | | 88.5% | | | 97.6% | | | 94.5% | |
| | Hypernetwork | 32.7% | 0.0% | 0.0% | 14.5% | 0.0% | 0.0% | 38.8% | 16.3% | 0.2% | 47.1% | 0.0% | 0.0% | 43.6% | 0.0% | 0.1% |
| | Naïve INR | 89.9% | 2.4% | 2.2% | 91.8% | 3.6% | 10.1% | 84.9% | 6.2% | 0.3% | 91.3% | 21.5% | 20.6% | 88.3% | 4.7% | 7.2% |
| | FD INR | 90.0% | 0.4% | 1.9% | 87.2% | 0.0% | 0.0% | 88.7% | 26.2% | 0.4% | 86.7% | 0.0% | 0.0% | 82.7% | 0.0% | 0.0% |
| | Naïve Emb. | 94.1% | 0.4% | 0.7% | – | – | – | – | – | – | 16.1% | 0.0% | – | – | – | – |
| | FD Emb. | 92.8% | 1.3% | 2.1% | – | – | – | – | – | – | 0.2% | 0.0% | – | – | – | – |
| **F-MNIST** | *Classifier* | | 79.8% | | | 73.3% | | | 79.8% | | | 83.2% | | | 82.6% | |
| | Hypernetwork | 37.0% | 0.0% | 0.1% | 21.0% | 13.0% | 15.5% | 34.9% | 13.0% | 0.8% | 82.8% | 5.0% | 21.5% | 33.0% | 18.6% | 30.4% |
| | Naïve INR | 79.8% | 5.8% | 17.9% | 79.0% | 8.5% | 20.8% | 80.6% | 3.1% | 5.7% | 95.7% | 16.1% | 0.0% | 81.8% | 5.2% | 13.0% |
| | FD INR | 81.5% | 4.4% | 13.9% | 80.8% | 5.2% | 15.2% | 73.6% | 0.9% | 0.3% | 91.9% | 0.2% | 0.0% | 75.0% | 1.7% | 10.4% |
| | Naïve Emb. | 80.8% | 5.7% | 21.6% | – | – | – | – | – | – | 80.1% | 1.0% | 9.4% | – | – | – |
| | FD Emb. | 81.3% | 10.5% | 27.2% | – | – | – | – | – | – | 82.8% | 5.0% | 21.5% | – | – | – |
| **CIFAR-10** | *Classifier* | | 45.1% | | | 40.2% | | | 40.6% | | | 48.2% | | | 48.2% | |
| | Hypernetwork | 33.7% | 0.1% | 0.5% | 15.5% | 0.3% | 0.4% | 39.2% | 0.0% | 0.4% | 28.2% | 0.1% | 0.2% | 33.4% | 0.4% | 1.1% |
| | Naïve INR | 41.9% | 0.1% | 0.5% | 45.3% | 1.7% | 3.7% | 47.9% | 2.7% | 5.0% | 49.3% | 4.7% | 4.1% | 46.3% | 3.0% | 4.0% |
| | FD INR | 48.7% | 2.6% | 5.2% | 45.7% | 1.1% | 2.8% | 45.1% | 0.1% | 0.2% | 46.0% | 0.1% | 0.2% | 43.2% | 0.0% | 0.3% |
| | Naïve Emb. | 47.6% | 2.4% | 4.6% | – | – | – | – | – | – | 49.3% | 4.7% | 0.8% | – | – | – |
| | FD Emb. | 42.5% | 9.0% | 13.6% | – | – | – | – | – | – | 46.0% | 0.1% | 2.1% | – | – | – |

*Table 25.* **Surrogate clean accuracy (%) for PGD steps ablation.** We report clean accuracy of each surrogate model. "–" indicates the surrogate is not applicable.

| Dataset | Surrogate | Inr2Vec | DWS | NFN | NFT | ScaleGMN |
|---|---|---|---|---|---|---|
| **MNIST** | Hypernetwork | 82.6% | 29.8% | 81.1% | 98.9% | 85.3% |
| | Naïve INR | 95.6% | 97.2% | 95.3% | 97.7% | 95.9% |
| | FD INR | 96.0% | 96.4% | 97.9% | 97.5% | 96.2% |
| | Naïve Emb. | 97.6% | – | – | 98.5% | – |
| | FD Emb. | 96.1% | – | – | 97.1% | – |
| **F-MNIST** | Hypernetwork | 82.2% | 25.7% | 86.3% | 91.7% | 51.5% |
| | Naïve INR | 87.8% | 88.7% | 85.7% | 88.4% | 88.6% |
| | FD INR | 87.2% | 88.6% | 86.3% | 88.8% | 82.2% |
| | Naïve Emb. | 88.7% | – | – | 84.4% | – |
| | FD Emb. | 84.7% | – | – | 86.5% | – |
| **CIFAR-10** | Hypernetwork | 45.2% | 20.4% | 51.2% | 68.7% | 41.9% |
| | Naïve INR | 49.5% | 50.3% | 51.7% | 52.1% | 51.6% |
| | FD INR | 51.0% | 49.4% | 52.0% | 52.6% | 49.8% |
| | Naïve Emb. | 51.5% | – | – | 47.3% | – |
| | FD Emb. | 44.1% | – | – | 47.9% | – |

*Table 26.* **Impact of PGD steps.** We report perturbed accuracy (%) under each step count. Default is 20 steps. "–" indicates the attack is not applicable.

| Dataset | Attack | Inr2Vec 5 | 10 | 20 | 30 | DWS 5 | 10 | 20 | 30 | NFN 5 | 10 | 20 | 30 | NFT 5 | 10 | 20 | 30 | ScaleGMN 5 | 10 | 20 | 30 |
|---|---|---|---|---|---|---|---|---|---|---|---|---|---|---|---|---|---|---|---|---|---|
| **MNIST** | *Classifier* | | 89.4% | | | | 75.5% | | | | 88.5% | | | | 97.6% | | | | 94.5% | | |
| | Hypernetwork | 0.1% | 0.0% | 0.0% | 0.0% | 0.1% | 0.0% | 0.0% | 0.0% | 1.6% | 0.4% | 0.2% | 0.1% | 0.0% | 0.0% | 0.0% | 0.0% | 0.5% | 0.1% | 0.1% | 0.1% |
| | Naïve INR | 7.3% | 2.1% | 2.2% | 2.2% | 9.7% | 8.4% | 10.1% | 10.5% | 1.3% | 0.4% | 0.3% | 0.3% | 14.8% | 16.9% | 20.6% | 21.5% | 8.6% | 5.6% | 7.2% | 9.4% |
| | FD INR | 2.7% | 1.8% | 1.9% | 1.9% | 0.9% | 0.0% | 0.0% | 0.0% | 0.4% | 0.4% | 0.4% | 0.4% | 0.0% | 0.0% | 0.0% | 0.0% | 0.0% | 0.0% | 0.0% | 0.0% |
| | Naïve Emb. | 2.9% | 0.8% | 0.7% | 0.8% | – | – | – | – | – | – | – | – | 0.2% | 0.0% | 0.0% | 0.0% | – | – | – | – |
| | FD Emb. | 1.9% | 1.5% | 2.0% | 2.4% | – | – | – | – | – | – | – | – | 0.0% | 0.0% | 0.0% | 0.0% | – | – | – | – |
| **F-MNIST** | *Classifier* | | 79.8% | | | | 73.3% | | | | 79.8% | | | | 83.2% | | | | 82.6% | | |
| | Hypernetwork | 0.4% | 0.2% | 0.1% | 0.2% | 15.9% | 15.6% | 15.5% | 15.5% | 2.2% | 1.3% | 0.8% | 0.5% | 2.1% | 1.4% | 0.8% | 0.5% | 30.9% | 30.6% | 30.4% | 30.4% |
| | Naïve INR | 20.9% | 18.5% | 17.9% | 18.2% | 22.4% | 19.6% | 20.8% | 21.6% | 8.6% | 7.8% | 7.5% | 7.6% | 13.7% | 12.5% | 12.4% | 12.3% | 14.5% | 13.2% | 13.0% | 13.1% |
| | FD INR | 15.6% | 14.4% | 13.9% | 13.9% | 15.7% | 15.1% | 15.2% | 15.1% | 2.0% | 0.9% | 0.3% | 0.1% | 5.2% | 4.3% | 3.8% | 3.7% | 11.8% | 10.9% | 10.4% | 10.1% |
| | Naïve Emb. | 25.4% | 22.5% | 21.6% | 21.4% | – | – | – | – | – | – | – | – | 10.5% | 9.6% | 9.4% | 9.3% | – | – | – | – |
| | FD Emb. | 28.1% | 27.4% | 27.2% | 27.2% | – | – | – | – | – | – | – | – | 23.0% | 21.9% | 21.5% | 21.5% | – | – | – | – |
| **CIFAR-10** | *Classifier* | | 45.1% | | | | 40.2% | | | | 40.6% | | | | 48.2% | | | | 48.2% | | |
| | Hypernetwork | 0.7% | 0.5% | 0.5% | 0.4% | 0.4% | 0.4% | 0.4% | 0.4% | 1.7% | 1.0% | 0.4% | 0.2% | 0.6% | 0.2% | 0.2% | 0.1% | 1.2% | 1.1% | 1.1% | 1.1% |
| | Naïve INR | 0.5% | 0.6% | 0.5% | 0.5% | 3.8% | 3.7% | 3.7% | 3.7% | 5.5% | 5.0% | 5.0% | 4.8% | 4.7% | 4.3% | 4.1% | 4.2% | 4.3% | 4.0% | 4.0% | 3.8% |
| | FD INR | 4.1% | 5.2% | 5.2% | 5.2% | 3.6% | 3.0% | 2.8% | 2.8% | 0.2% | 0.2% | 0.2% | 0.1% | 0.3% | 0.2% | 0.2% | 0.2% | 0.3% | 0.3% | 0.3% | 0.3% |
| | Naïve Emb. | 5.5% | 4.8% | 4.6% | 4.6% | – | – | – | – | – | – | – | – | 1.0% | 0.8% | 0.8% | 0.8% | – | – | – | – |
| | FD Emb. | 13.8% | 13.5% | 13.4% | 13.6% | – | – | – | – | – | – | – | – | 2.4% | 2.2% | 2.1% | 2.0% | – | – | – | – |

## C.7. Impact on Attack Surface

Table 27 compares input-, function-, and embedding-space attacks across all pipelines. Table 28 reports end-to-end attack results on MWT.

*Table 27*. **Attack surface comparison across pipelines.** We report classifier clean accuracy and perturbed accuracy (%) for input-, function-, and embedding-space attacks. "–" indicates the attack is not applicable.

| Dataset | Attack Surface | Clean Accuracy | | | | | Perturbed Accuracy | | | | |
|---|---|---|---|---|---|---|---|---|---|---|---|
| | | Inr2Vec | DWS | NFN | NFT | ScaleGMN | Inr2Vec | DWS | NFN | NFT | ScaleGMN |
| **MNIST** | Input | 88.99% | 75.48% | 90.76% | 98.36% | 94.75% | 0.01% | 0.01% | 0.17% | 0.00% | 0.00% |
| | Function | | | | | | 5.09% | 47.74% | 25.80% | 16.18% | 4.39% |
| | Embedding | | | | | | 0.00% | – | – | 0.00% | – |
| **F-MNIST** | Input | 78.67% | 74.63% | 79.27% | 83.79% | 82.78% | 0.13% | 15.15% | 0.27% | 0.78% | 10.44% |
| | Function | | | | | | 46.68% | 68.39% | 64.03% | 55.97% | 48.41% |
| | Embedding | | | | | | 8.72% | – | – | 0.00% | – |
| **CIFAR-10** | Input | 35.13% | 39.63% | 41.50% | 48.87% | 48.56% | 0.46% | 0.38% | 0.15% | 0.15% | 0.25% |
| | Function | | | | | | 30.78% | 38.52% | 34.59% | 39.82% | 32.37% |
| | Embedding | | | | | | 7.01% | – | – | 0.00% | – |

*Table 28*. **End-to-end attack on MWT.** We report clean and perturbed accuracy (%) under input-space PGD attack.

| Dataset | Clean | Perturbed |
|---|---|---|
| **MNIST** | 98.50% | 84.10% |
| **F-MNIST** | 89.50% | 59.30% |
| **CIFAR-10** | 56.80% | 1.60% |

## C.8. Additional Attack Methods

Beyond PGD, we evaluate three additional attacks of distinct paradigms: a function-space Square Attack (gradient-free and query-based), DeepFool (first-order and minimum-perturbation), and a One-Pixel attack (gradient-free black-box).

**Function-space Square Attack** We additionally develop a function-space variant of Square Attack (Andriushchenko et al., 2020), a gradient-free, query-based attack. It perturbs random subsets of the INR weight parameters under the same image-space constraint as our function-space PGD, requiring only query access to model predictions.

*Table 29*. **Function-space Square Attack on MNIST.** Gradient-free, black-box; clean and perturbed accuracy (%) with 1,000 queries.

| Dataset | Classifier | Clean | Square Attack |
|---|---|---|---|
| | **Inr2Vec** | 88.99% | 10.11% |
| | **DWS** | 75.48% | 37.84% |
| **MNIST** | **NFN** | 90.76% | 13.17% |
| | **NFT** | 98.36% | 73.66% |
| | **ScaleGMN** | 94.75% | 82.85% |

Table 29 shows this gradient-free attack already degrades the more vulnerable pipelines substantially (Inr2Vec to $10.11\%$, NFN to $13.17\%$), while NFT and ScaleGMN remain far more resilient ($73.66\%$ and $82.85\%$), under this gradient-free function-space attack. Even so, our input-space surrogate attacks remain markedly stronger than any function-space attack, reinforcing that input-space vulnerabilities are the more critical surface.

**DeepFool and One-Pixel** We evaluate two further attacks from distinct paradigms. DeepFool (Moosavi-Dezfooli et al., 2016) is a first-order attack that seeks the minimal perturbation crossing the decision boundary; since it requires gradients through the INR fitting process, we apply it through our surrogate framework. The One-Pixel attack (Su et al., 2019) is a gradient-free, differential-evolution black-box attack; being query-only, it directly queries the INR pipeline without surrogate approximation.

*Table 30.* **DeepFool on MNIST.** Surrogate clean and perturbed (Adv.) accuracy (%) under each INR surrogate.

| Dataset | Classifier | Naïve INR | | FD INR | |
|---|---|---|---|---|---|
| | | Clean | Adv. | Clean | Adv. |
| | **Inr2Vec** | 95.6% | 9.0% | 96.0% | 8.9% |
| | **DWS** | 97.2% | 40.8% | 96.4% | 35.3% |
| **MNIST** | **NFN** | 95.3% | 28.5% | 97.9% | 32.7% |
| | **NFT** | 97.7% | 40.7% | 97.5% | 33.8% |
| | **ScaleGMN** | 95.9% | 43.1% | 96.2% | 34.5% |

*Table 31.* **One-Pixel attack on MNIST.** Perturbed accuracy (%) under differential evolution on 1,000 randomly sampled test samples.

| Dataset | Classifier | Cls. Acc | 1-pixel | 3-pixel | 5-pixel |
|---|---|---|---|---|---|
| | **Inr2Vec** | 89.0% | 46.6% | 40.4% | 35.7% |
| | **DWS** | 75.5% | 69.4% | 64.9% | 61.8% |
| **MNIST** | **NFN** | 90.8% | 73.6% | 68.6% | 66.2% |
| | **NFT** | 98.4% | 95.1% | 93.8% | 89.6% |
| | **ScaleGMN** | 94.8% | 91.7% | 88.6% | 87.1% |

Through the surrogate, DeepFool reduces accuracy to roughly 9% for Inr2Vec and to 28–43% for the other classifiers, down from 95–98% surrogate clean accuracy, confirming vulnerability even under minimal perturbations (Table 30). In contrast, One-Pixel causes only moderate degradation on Inr2Vec and is largely ineffective elsewhere (Table 31); increasing the number of perturbed pixels ($1 \rightarrow 3 \rightarrow 5$) strengthens it monotonically but accuracy remains far above that of our surrogate-based first-order attacks. This gap between first-order and gradient-free attacks underscores why a surrogate framework is necessary for effective first-order auditing of INR-based classifiers.

### C.9. Additional results on Improving Robustness on INR-based Classifiers

**Adversarial training.** Table 32 shows the effect of end-to-end adversarial training on MWT. Table 33 reports embedding-space adversarial training results for encoder-based pipelines.

*Table 32.* **Effect of end-to-end adversarial training on MWT.** We report clean and perturbed accuracy (%) before and after adversarial training (AT).

| Dataset | Clean | Pert. | Adv. Clean | Adv. Pert. |
|---|---|---|---|---|
| **MNIST** | 98.50% | 84.10% | 97.96% | 86.85% |
| **F-MNIST** | 89.90% | 57.92% | 88.87% | 51.10% |
| **CIFAR-10** | 56.80% | 1.60% | 55.98% | 15.36% |

*Table 33.* **Effect of embedding-space adversarial training.** We report clean and perturbed accuracy (%) before and after adversarial training (AT) for encoder-based pipelines.

| Dataset | Inr2Vec | | | | NFT | | | |
|---|---|---|---|---|---|---|---|---|
| | Clean | Pert. | Adv Clean | Adv Pert. | Clean | Pert. | Adv Clean | Adv Pert. |
| **MNIST** | 88.99% | 0.00% | 86.79% | 0.00% | 98.36% | 0.00% | 97.52% | 11.35% |
| **F-MNIST** | 78.67% | 8.72% | 77.18% | 43.44% | 83.79% | 0.00% | 81.79% | 51.09% |
| **CIFAR-10** | 35.13% | 7.01% | 40.90% | 11.80% | 48.87% | 0.00% | 43.48% | 1.50% |

**An INR-specific regularization.** Existing defenses are designed for discrete inputs; here we consider a candidate tailored to the INR pipeline. To the best of our knowledge, no defense has been specifically designed for INR-based classifiers. A natural starting point is to regularize the INR fitting objective so that the fitted weights are less able to overfit adversarial

perturbations. We add an $L_2$ penalty to the fitting objective (Equation (1)):

$$\theta^*(x) = \arg\min_{\theta} \ \|f_\theta(G) - x\|^2 + \lambda_{\text{reg}} \|\theta\|^2,$$ (2)

and train NFN on MNIST with varying $\lambda_{\text{reg}}$ (Table 34).

*Table 34.* **Effect of $L_2$ regularization on INR fitting.** We train NFN with varying $\lambda_{\text{reg}}$ in the INR fitting objective on MNIST. We report classifier clean accuracy and surrogate clean and perturbed accuracy (%).

| $\lambda_{\text{reg}}$ | *Classifier* | Naïve INR Surr. | | FD INR Surr. | |
|---|---|---|---|---|---|
| | | **Clean** | **Pert.** | **Clean** | **Pert.** |
| 0 **(no reg)** | 90.76% | 95.27% | 0.29% | 97.85% | 0.40% |
| $10^{-4}$ | 89.39% | 96.43% | 0.01% | 95.45% | 0.12% |
| $10^{-3}$ | 26.21% | 96.41% | 0.00% | 95.32% | 0.19% |
| $10^{-2}$ | 10.07% | 96.95% | 0.00% | 96.92% | 0.23% |

We observe no meaningful robustness benefit: under both Naïve and FD INR surrogate attacks, perturbed accuracy remains near zero across all regularization strengths. Moreover, for $\lambda_{\text{reg}} \geq 10^{-3}$ the classifier's clean accuracy collapses (to $26.2\%$ and $10.1\%$), making these settings impractical irrespective of any robustness effect. Thus, simply constraining the INR fitting objective does not mitigate the representation-specific vulnerability, and effective INR-specific defenses remain open.

## D. Auditing via the Classifier Pipeline under Known Initialization

**Setup.** In the main text, surrogate-crafted adversarial examples are evaluated on the surrogate pipeline because the INR-generation process is non-differentiable and the fitting initialization is assumed unknown (§3.1). As a complementary analysis, we consider a stronger-information setting in which the auditor additionally assumes access to the fitting initialization. This allows us to test how much of the surrogate-crafted perturbation survives an additional INR refitting step: we refit an INR to the perturbed image using the known initialization and then pass the refitted INR through the downstream classifier. Table 35 reports clean and perturbed accuracy under this setting for all three datasets, five classifiers, and surrogate variants.

**Observation.** The additional refitting step substantially changes how surrogate-crafted perturbations affect the classifier pipeline. While the surrogate-pipeline evaluation in the main text yields large accuracy drops (Tables 1 and 2), the classifier pipeline exhibits substantial variation across classifiers and datasets (Table 35). On MNIST, worst-case accuracy ranges from $\sim 22\%$ for DWS to $\sim 74\%$ for ScaleGMN. On CIFAR-10, the additional refitting step removes most of the surrogate-crafted signal, with perturbed accuracy remaining close to clean accuracy for the evaluated classifiers. The effect is also attack-surface dependent: for encoder-based pipelines, embedding-space surrogates are markedly stronger than INR-space ones (e.g., NFT on MNIST: $48.4\%$ under FD Embedding vs. $80.7\%$ under FD INR). Overall, this setting suggests that refitting can dilute a large fraction of the surrogate-crafted signal, and that the degree of dilution depends on the classifier structure. We attribute this to two factors.

- **Factor 1: richness of built-in weight-space invariance.** Refitting from a fixed initialization still admits variation along permutation, scaling, and other nuisance directions in weight space. The five classifiers differ in how much of this variation they are invariant to: Inr2Vec has no built-in equivariance, DWS and NFN enforce permutation equivariance (the latter through structured functional layers), NFT uses attention-based weight-space invariance, and ScaleGMN adds scaling equivariance on top. The robustness ordering on the classifier pipeline broadly follows this ordering, consistent with the equivariance analysis in §4.3: models invariant to a larger nuisance subspace project out more of the refit-induced adversarial variation.

- **Factor 2: two-stage frozen encoder.** Inr2Vec and NFT are encoder-based: their encoder is trained with a non-classification objective, frozen, and only then followed by a classifier. The frozen encoder acts as a fixed bottleneck that further reduces surrogate perturbations passing through INR-space attacks—explaining Inr2Vec's robustness under INR-space surrogates despite its weak invariance. The protection is asymmetric: embedding-space surrogates approximate the encoder output directly and bypass the bottleneck, which is why encoder-based pipelines are most vulnerable under embedding-space surrogates ($90\% \rightarrow 38\%$ for Inr2Vec; $99\% \rightarrow 48\%$ for NFT on MNIST), consistent with our threat model treating these as distinct surrogate surfaces (§3.1).

*Table 35*. **Auditing via the classifier pipeline under known initialization.** Clean and perturbed accuracy (%) when surrogate-crafted adversarial examples are refit into an INR with the training-time initialization and passed through the downstream classifier. Gaussian and Transfer are non-surrogate baselines; the remaining rows are our INR-space (INR) and embedding-space (Emb.) surrogates. "–" indicates the surrogate is not applicable to that classifier.

| | Attack | Inr2Vec | DWS | NFN | NFT | ScaleGMN |
|---|---|---|---|---|---|---|
| | *Classifier* | 89.6% | 88.5% | 92.2% | 99.0% | 97.7% |
| **MNIST** | Gaussian | 79.6% | 14.9% | 35.4% | 86.4% | 76.3% |
| | Transfer | 82.6% | 32.2% | 72.9% | 95.4% | 79.7% |
| | Hypernetwork | 74.7% | 30.2% | 78.4% | 91.0% | 78.8% |
| | Naïve INR | 65.3% | 24.0% | 67.4% | 82.3% | 75.6% |
| | FD INR | 62.3% | 21.6% | 78.9% | 80.7% | 73.8% |
| | Naïve Emb. | 57.3% | – | – | 67.5% | – |
| | FD Emb. | 38.2% | – | – | 48.4% | – |
| | *Classifier* | 78.6% | 70.7% | 80.1% | 84.5% | 83.9% |
| **Fashion-MNIST** | Gaussian | 76.2% | 51.5% | 74.5% | 81.1% | 76.2% |
| | Transfer | 77.0% | 63.0% | 76.8% | 81.8% | 78.5% |
| | Hypernetwork | 76.4% | 62.2% | 76.1% | 81.0% | 77.1% |
| | Naïve INR | 72.4% | 58.9% | 70.8% | 74.9% | 77.0% |
| | FD INR | 71.1% | 58.9% | 70.6% | 77.2% | 75.7% |
| | Naïve Emb. | 70.8% | – | – | 67.8% | – |
| | FD Emb. | 69.3% | – | – | 61.6% | – |
| | *Classifier* | 38.3% | 36.6% | 47.6% | 57.8% | 56.0% |
| **CIFAR-10** | Gaussian | 33.1% | 36.0% | 40.7% | 55.1% | 54.4% |
| | Transfer | 34.0% | 36.0% | 43.7% | 56.4% | 53.4% |
| | Hypernetwork | 34.0% | 35.0% | 43.5% | 54.5% | 52.6% |
| | Naïve INR | 33.9% | 36.5% | 42.1% | 49.5% | 52.9% |
| | FD INR | 32.8% | 37.5% | 42.7% | 49.9% | 50.5% |
| | Naïve Emb. | 32.8% | – | – | 47.2% | – |
| | FD Emb. | 32.7% | – | – | 44.6% | – |

**Interpretation.** These results should be interpreted as a complementary refitting analysis rather than as the primary auditing setting. First, for the direct pipelines DWS and NFN on MNIST, random Gaussian noise also causes sizable accuracy drops (DWS: $88\% \to 15\%$; NFN: $92\% \to 35\%$), indicating that part of the drop in the classifier pipeline may reflect general input sensitivity in these models. Second, Factors 1 and 2 provide a structural explanation for the observed trends, but we leave direct geometric or spectral characterization of the refit-induced variation to future work. Under the default threat model with unknown initialization (§3.1), the surrogate pipeline is the intended auditing interface used in the main text. This complementary setting instead measures how much of the surrogate-crafted signal survives an additional INR refitting step.

