# OpenReview forum: "Adversarial Robustness of Implicit Neural Representation-Based Classifiers"
_ICML.cc/2026/Conference — ICML 2026 regular_

### Official Review · Reviewer_uanU · 2026-03-03

**Soundness:** 4
**Presentation:** 4
**Significance:** 4
**Originality:** 3
**Overall Recommendation:** 5
**Confidence:** 4

**Summary:**

The paper presents a thorough investigation of the adversarial robustness of classifiers based on Implicit Neural Representations (INR). In contrast to prior work, it shows that INRs are not more robust than direct classification methods. While the inherent loopy optimization of INRs might be able to obscure gradient based attacks, the authors show that this can be overcome via surrogates using non-intrusive gradient estimations.
Evaluating a wide range of settings including different INR architectures, surrogates, attack methods and datasets, the paper is able to conclusively show that INRs can be attacked like any other classification architecture, throughout dropping classification accuracy close to zero. An additional investigation of common defenses  also show limited effects.

**Compliance With Llm Reviewing Policy:**

Affirmed.

**Final Justification:**

The rebuttal phase helped to clarify many of the open points/questions by the other reviewers. Hence,  I'm confident to keep my accept score.

**Key Questions For Authors:**

Following up on the weakness section, I was wondering if regularization of eq. 1 might increase robustness. Limiting the over fitting of the INR might make it harder for perturbations... do you have any thoughts/results on this?

**Limitations:**

yes

**Strengths And Weaknesses:**

**Strengths**
The paper is well structured, easy to follow, discusses the relevant related work and tackles an important topic, reversing key assumptions made in current literature. Hence it represents an important contribution to field.

At a technical level, the paper utilizes known methods (non-intrusive gradient estimation) to solve an so far unsolved problem. The applied methods themselves are technically sound and well formulated.

The experimental evaluation is excessive, covering a wide range of the possible evaluation space. The obtained results are very clear, leaving no doubts in the claimed reversal of the results of prior works.

The authors provide the relevant code, making the results easy to reproduce.

**Weaknesses**
 Searching for weaknesses, 4.4 is somewhat limited to standard robustness methods (denoising and adversarial training), not discussing the possibility INR specific defenses. Are there any in literature? Are there possible approaches to try?

---

> ### Author Rebuttal · Authors · 2026-03-31
>
> **General Response** We thank you for your thoughtful feedback. We address each concern and provide new experimental results that strengthen our findings. Our key additions include:
> - Table A: **Extension to 3D point cloud classification on ShapeNet-10**
> - Table B: **Function-space AutoAttack** for AT evaluation
> - Table C: **FD supervision ablation** on gradient approximation quality
> - Table D: **Function-space Square Attack** as a gradient-free black-box baseline
> - Table E: **L2 regularization** as an INR-specific defense
>
> **W1. No discussion of INR-specific defenses**
>
> To the best of our knowledge, no defense mechanisms have been specifically designed for INR-based classifiers. We investigate one natural candidate -- L2 regularization of the INR fitting objective -- as suggested by the reviewer.
>
>
> **Q1. Would regularization of INR fitting (Eq. 1) increase robustness?**
>
> We thank the reviewer for this insightful suggestion. We conducted additional experiments on MNIST with L2 regularization applied to the INR fitting objective:
> $$ \theta^*(x) = \underset{\theta}{argmin} \| f_\theta(\mathcal{G}) - x\|^2 + \lambda_\text{reg} \cdot \|\theta\|^2. $$
>
> >**Table E. Effect of L2 regularization on INR fitting. We train NFN with varying $\lambda_\text{reg}$ in INR fitting on MNIST.**
> >| $\lambda_\text{reg}$ | Cls Clean | Naïve INR Surr. Clean | Naïve INR Surr. Pert. | FD INR Surr. Clean | FD INR Surr. Pert. |
> >|-------|-----------|------------|----------|---------|--------|
> >| 0 (no reg) | 90.76% | 95.27% | 0.29% | 97.85% | 0.40% |
> >| 1e-4  | 89.39% | 96.43% | 0.01% | 95.45% | 0.12% |
> >| 1e-3  | 26.21% | 96.41% | 0.00% | 95.32% | 0.19% |
> >| 1e-2  | 10.07% | 96.95% | 0.00% | 96.92% | 0.23% |
>
> At $\lambda_\text{reg} \ge 1e-03$, classifier performance collapses to 26.2% and 10.1%, making these configurations impractical regardless of their robustness properties. Setting that aside, adversarial accuracy remains near zero across all regularization strengths and both surrogate types with minor fluctuations, indicating that L2 regularization provides no meaningful robustness improvement.

---

> > ### Author Rebuttal · Reviewer_uanU · 2026-04-02
> >
> > Thank you for your answers which resolve my questions. I can partially follow some of the arguments made by the other reviewers and it will up to further discussions to see if their concerns have been met. Do far I'm maintaining my score.

---

### Official Review · Reviewer_ddh3 · 2026-03-12

**Soundness:** 2
**Presentation:** 2
**Significance:** 2
**Originality:** 3
**Overall Recommendation:** 3
**Confidence:** 3

**Summary:**

This paper investigates the adversarial robustness of classifiers built on implicit neural representations (INRs). Because INR-based pipelines require solving an optimization problem for each input image, standard gradient-based attacks are computationally infeasible. The authors address this challenge by introducing surrogate models that approximate the INR generation process and enable first-order adversarial attacks. Experiments on MNIST, Fashion-MNIST, and CIFAR-10 across multiple INR classification pipelines demonstrate that these models are highly vulnerable to adversarial perturbations once proper attacks are applied. The study further analyzes architectural factors influencing robustness and evaluates existing defenses, showing that they provide limited protection. The results suggest that INR-based classifiers are not inherently robust and expose representation-specific vulnerabilities.

**Compliance With Llm Reviewing Policy:**

Affirmed.

**Key Questions For Authors:**

Please see the weakness for the questions.

**Limitations:**

yes

**Strengths And Weaknesses:**

Strengths

1. The authors conduct a systematic study of adversarial robustness in INR-based classifiers.
2. The paper presents a surrogate-based robustness auditing framework for INR-based classifiers.

Weaknesses

1. The presentation of the paper is somewhat confusing. I am not entirely sure what the authors are primarily focusing on, and the contributions are not very clear. I suggest that the authors summarize their contributions in bullet-point form so that they are clearer and easier to understand.

2. In the experimental section, it appears that there are no experiments directly evaluating PGD attacks. If that is the case, it is unclear why PGD attacks are introduced and discussed in the related work.

3. In Table 1, only the results of the best surrogate are reported, and it is indicated that the best surrogate achieves significantly better performance than other methods. Have the authors evaluated other black-box attack methods? Additionally, it would be helpful to repeat the experiments multiple times to reduce the impact of stochastic effects.

4. Although the authors attempt to cover many details of the proposed method, the presentation becomes somewhat overwhelming and makes it difficult to follow the main idea of the paper.

---

> ### Author Rebuttal · Authors · 2026-03-31
>
> **General Response** We thank you for your thoughtful feedback. We address each concern and provide new experimental results that strengthen our findings. Our key additions include:
> - Table A: **Extension to 3D point cloud classification on ShapeNet-10**
> - Table B: **Function-space AutoAttack** for AT evaluation
> - Table C: **FD supervision ablation** on gradient approximation quality
> - Table D: **Function-space Square Attack** as a gradient-free black-box baseline
> - Table E: **L2 regularization** as an INR-specific defense
>
> **W1. Contributions are not clear**
>
> We will restructure the introduction to clearly list our contributions:
> - We present the **first systematic study** of adversarial robustness in INR-based classifiers, evaluating six classifiers across three pipeline designs.
> - We introduce a **surrogate-based auditing framework** that overcomes the computational challenges of differentiating through INR fitting, enabling standard first-order attacks in INR-based classifiers.
> - We demonstrate that **INR-based classifiers are not inherently robust**, contrary to prior claims, with accuracy collapsing to near zero under our attacks.
> - We evaluate **existing defenses** and show they provide limited protection for INR-based classifiers.
>
> **W2. No experiments directly evaluating PGD attacks**
>
> We clarify that all our attacks use PGD as the attack algorithm. Specifically, our surrogate-based attacks (Hypernetwork, Naïve/FD INR, Naïve/FD Embedding surrogate) all employ 20-step PGD using gradients obtained from the trained surrogates (Section 4.1). The surrogates provide the gradient approximation; PGD is the attack algorithm applied using those gradients. Additionally, our function-space attacks (Table 5) directly apply PGD in INR weight space.
>
>
> **W3. Best surrogate reporting, other black-box attacks, and experiment repetition**
>
> Table 1 reports the best surrogate for conciseness, while Table 2 provides a full comparison across all five surrogate types for each classifier and dataset.
> Regarding additional black-box attacks, our transfer attack in Table 1 (using ResNet-18 as a surrogate) serves as a black-box attack. In this rebuttal, we additionally develop a function-space variant of Square Attack (Andriushchenko et al., 2020), a gradient-free, query-based attack that perturbs random subsets of INR weight parameters under the same image-space constraint as our function-space PGD.
>
> > **Table D. function-space Square Attack on MNIST. We report clean accuracy and perturbed accuracy (%) with 1000 queries.**
> >| Classifier | Clean  | function-space Square Attack   |
> >|-----------|--------|---------|
> >| DWS       | 75.48% | 37.84%  |
> >| Inr2Vec   | 88.99% | 10.11%  |
> >| NFN       | 90.76% | 13.17%  |
> >| NFT       | 98.36% | 73.66%  |
> >| ScaleGMN  | 94.75% | 82.85%  |
>
> Square Attack achieves significant accuracy drops across classifiers, reducing Inr2Vec to 10.11% and NFN to 13.17% with 1000 queries -- surpassing function-space PGD (5.09% and 25.80% in Table 5, respectively) without requiring any gradient information. NFT and ScaleGMN show more resilience (73.66% and 82.85%), consistent with their higher robustness observed across other attack methods. Nonetheless, input-space surrogate attacks (near 0%, Table 1) remain substantially more effective than any function-space attack, reinforcing that input-space vulnerabilities are more critical.
>
> Regarding experiment repetition, we will consider including variance analysis in the camera-ready version.
>
> **W4. Too much detail makes main idea hard to follow**
>
> We appreciate this feedback. We will restructure the paper in the camera-ready version, moving secondary analyses to the appendix to improve readability while keeping the main narrative focused.
>
> **References**
> - Andriushchenko, M. et al., Square Attack: a query-efficient black-box adversarial attack via random search. ECCV 2020.

---

> > ### Author Rebuttal · Reviewer_ddh3 · 2026-04-02
> >
> > Thanks for your rebuttal. It makes things clearer now.
> >
> > I have another question. Have you tried different attack methods to evaluate the performance of each model, such as Jitter, DeepFool, One-Pixel, or other state-of-the-art attacks? Have you also tried increasing the attack intensity to observe how the model responds?

---

> > > ### Author Response · Authors · 2026-04-06
> > >
> > > We thank the reviewer for engaging with us and for the follow-up questions.
> > >
> > > ----
> > >
> > > **FQ1. Different attack methods.**
> > >
> > > We evaluate two additional attack methods: DeepFool (Moosavi-Dezfooli et al., 2016), a first-order attack that finds minimal perturbations crossing the decision boundary, and One-Pixel attack (Su et al., 2019), a gradient-free black-box attack based on differential evolution. These two methods represent fundamentally different attack paradigms -- first-order gradient-based and gradient-free -- providing complementary perspectives on model vulnerability. We note that first-order methods such as PGD and DeepFool require gradient computation through the INR fitting process, which is intractable due to the optimization-in-the-loop structure; we therefore apply them through our surrogate framework. One-Pixel attack, being gradient-free and requiring only query access to the model's predictions, directly queries the INR pipeline without surrogate approximation.
> > >
> > >
> > > >**Table H. DeepFool on MNIST. We report surrogate clean and perturbed accuracy (%).**
> > > >|Classifier|Naïve INR Clean|Naïve INR Adv|FD INR Clean|FD INR  Adv|
> > > >|---|---|---|---|---|
> > > >|Inr2Vec|95.6|9.0|96.0|8.9|
> > > >|DWS|97.2|40.8|96.4|35.3|
> > > >|NFN|95.3|28.5|97.9|32.7|
> > > >|NFT|97.7|40.7|97.5|33.8|
> > > >|ScaleGMN|95.9|43.1|96.2|34.5|
> > >
> > > DeepFool substantially reduces accuracy across all classifiers, requiring only small perturbations (mean $L_2$ of 1.0-7.7), to cross the decision boundary, confirming that INR-based classifiers are vulnerable even under minimal perturbations.
> > >
> > >
> > > >**Table I. One-Pixel attack on MNIST (1,000 test samples). We report perturbed accuracy under differential evolution with 50 generations and population size 400.**
> > > >|Classifier|Classifier Acc|1-pixel|3-pixel|5-pixel|
> > > >|---|---|---|---|---|
> > > >|Inr2Vec|89.0|46.6|40.4|35.7|
> > > >|DWS|75.5|69.4|64.9|61.8|
> > > >|NFN|90.8|73.6|68.6|66.2|
> > > >|NFT|98.4|95.1|93.8|89.6|
> > > >|ScaleGMN|94.8|91.7|88.6|87.1|
> > >
> > > One-Pixel achieves moderate accuracy degradation on Inr2Vec but is largely ineffective against others. The results reveal a clear gap between first-order and gradient-free attacks: first-order methods reduce accuracy to near zero, while the gradient-free One-Pixel attack achieves only limited degradation. Since first-order methods require gradient computation through the INR fitting process which is intractable without approximation, our surrogate framework is essential for enabling these effective attacks and for properly auditing the robustness of INR-based classifiers.
> > >
> > > **FQ2. Attack intensity.**
> > >
> > > DeepFool finds the **minimal perturbation** that crosses the decision boundary by design; the notion of attack intensity is inherently undefined for minimum-perturbation attacks. For One-Pixel attack, we vary the number of perturbed pixels (1, 3, 5) to increase attack strength. As shown in Table I, increasing the number of pixels leads to stronger attacks, consistent with the trend observed in Section 4.3 where larger perturbation bound and more  PGD steps result in greater accuracy drops. Nevertheless, accuracy remains substantially higher than that achieved by our surrogate-based first-order attacks.
> > >
> > > **References**
> > > - Moosavi-Dezfooli, S.-M. et al., DeepFool: a simple and accurate method to fool deep neural networks. CVPR 2016.
> > > - Su, J. et al., One pixel attack for fooling deep neural networks. IEEE Transactions on Evolutionary Computation 2019.

---

### Official Review · Reviewer_MiDQ · 2026-03-13

**Soundness:** 3
**Presentation:** 3
**Significance:** 3
**Originality:** 3
**Overall Recommendation:** 4
**Confidence:** 3

**Summary:**

This paper investigates the robustness of image classifiers based on Implicit Neural Representation (INR) against adversarial attacks. The inclusion of an optimization-in-the-loop process between input and prediction makes standard first-order input attacks difficult to apply to INR, leading many to assume inherent robustness. The authors propose a surrogate-based auditing method. By training a differentiable surrogate to approximate either the INR generation process or the embedding directly, they enable attacks like PGD to be applied to INR. Experiments evaluate six representative INR classification pipelines on MNIST, Fashion-MNIST, and CIFAR-10. The experimental results support the paper's claims. The experiments further compare the effectiveness of different surrogates, analyze the transferability across pipelines, and examine how INR architecture and attack configurations affect robustness. They also test defense methods like diffusion denoising and adversarial training, finding that these generally provide only limited protection.

**Compliance With Llm Reviewing Policy:**

Affirmed.

**Key Questions For Authors:**

Please see Weaknesses

**Limitations:**

yes

**Strengths And Weaknesses:**

Strengths

1. The paper challenges the intuition that INR is inherently robust, demonstrating that suitable attack methods were previously unavailable. It employs a clever and simple approach to enable standard first-order attacks like PGD.

2. The experimental results are compelling. The paper shows that conventional transfer attacks are largely ineffective, while surrogate-based attacks significantly reduce accuracy, strongly supporting the core claim that “previous robustness conclusions may have been influenced by gradient masking.”

3. The ablation studies and analysis are relatively comprehensive. The authors compare different surrogate forms and analyze the impact of architecture, attack configuration, and defense methods.

Weaknesses

1. While the surrogate approach solves the problem of directly attacking INR fitting being infeasible, I am concerned about the cost of training the surrogate. Is this merely shifting an infeasible problem to one solvable at an acceptable cost? If the actual training cost of the surrogate is high, this seems like a transfer of the problem.

2. Table 1 states “All perturbed samples are evaluated on the surrogate pipeline.” Taken literally, this implies all attack samples are validated on the surrogate rather than the actual INR. While the surrogate's results are trained from INR parameters, we cannot ascertain how closely they approximate the real INR.

---

> ### Author Rebuttal · Authors · 2026-03-31
>
> **General Response** We thank you for your thoughtful feedback. We address each concern and provide new experimental results that strengthen our findings. Our key additions include:
> - Table A: **Extension to 3D point cloud classification on ShapeNet-10**
> - Table B: **Function-space AutoAttack** for AT evaluation
> - Table C: **FD supervision ablation** on gradient approximation quality
> - Table D: **Function-space Square Attack** as a gradient-free black-box baseline
> - Table E: **L2 regularization** as an INR-specific defense
>
> **W1. Surrogate training cost**
>
> We thank the reviewer for this question. We would like to clarify that differentiating through INR fitting is not merely expensive but effectively intractable in practice, as it requires unrolling thousands of optimization steps and storing intermediate states. As such, there is no practical baseline that enables first-order attacks in this setting.
> Our surrogate therefore does not transfer the computational burden of an existing feasible method; rather, it provides a tractable approximation that makes robustness evaluation possible in the first place. Moreover, the cost of training the surrogate is amortized, as it is incurred once and reused across all samples and attack iterations.
> We also emphasize that some surrogate approaches (for example, hypernetwork-based ones) are not introduced by us to bypass this issue, but are standard and widely used alternatives to per-instance INR training, alongside meta-learning approaches. Our goal is to be comprehensive and evaluate robustness across these different INR training regimes, rather than to propose a more efficient training method.
>
> **W2. How closely the surrogate approximates the real INR**
>
> We appreciate this concern. We provide two lines of evidence that our surrogate evaluation faithfully reflects the real pipeline's vulnerability:
> - Our surrogates achieve 95-98% clean accuracy on MNIST, closely matching the real classifiers' predictions, demonstrating that the surrogates' forward behavior is a faithful approximation.
> - The FD surrogate's loss $$L = \lambda_\text{rec} \cdot \| \hat{\theta}(x) - \theta^*(x)\|^2 + \lambda_\text{cls}  \cdot \text{CE}(C(\hat{\theta}(x)), y)+ \lambda_\text{der} \cdot \|\frac{\partial \hat{\theta}}{\partial x} \cdot \Delta x - \Delta \theta \|^2$$ includes a term (weighted by  $\lambda_\text{der}$) that directly supervises the surrogate's Jacobian to match the true INR fitting Jacobian. By varying $\lambda_\text{der}$ while fixing other loss weights, we isolate the effect of gradient approximation quality. To directly examine how gradient approximation quality affects attack stength, therefore, we ablate $\lambda_\text{der}$ (the weight of finite-difference supervision) on MNIST across all five classifiers.
>     >**Table C. Effect of finite-difference supervision strength on attack effectiveness on MNIST. Lower Jacobian loss indicates better gradient alignment with the true INR fitting process.**
>     >| Classifier | λ_der | Surr Clean | Jac. Loss | Adv Acc |
>     >|-----------|-------|-----------|-----------|---------|
>     >| NFT       | 0.01  | 97.85%    | 0.357     | 16.62%  |
>     >|           | 0.1   | 97.46%    | 0.345     | 0.98%   |
>     >|           | 1.0   | 97.86%    | 0.336     | 0.17%   |
>     >|           | 10.0  | 97.97%    | 0.334     | 0.12%   |
>     >| ScaleGMN  | 0.01  | 96.78%    | 0.417     | 22.83%  |
>     >|           | 0.1   | 97.89%    | 0.348     | 19.14%  |
>     >|           | 1.0   | 97.65%    | 0.335     | 12.75%  |
>     >|           | 10.0  | 98.22%    | 0.333     | 0.07%   |
>     >| DWS       | 0.01  | 96.77%    | 0.432     | 4.44%   |
>     >|           | 0.1   | 97.63%    | 0.360     | 7.50%   |
>     >|           | 1.0   | 97.09%    | 0.337     | 0.11%   |
>     >|           | 10.0  | 97.83%    | 0.334     | 0.00%   |
>     >| Inr2Vec   | 0.01  | 97.05%    | 0.403     | 0.02%   |
>     >|           | 0.1   | 97.38%    | 0.350     | 0.03%   |
>     >|           | 1.0   | 97.85%    | 0.337     | 0.00%   |
>     >|           | 10.0  | 97.88%    | 0.333     | 0.00%   |
>     >| NFN       | 0.01  | 95.78%    | 0.408     | 0.03%   |
>     >|           | 0.1   | 94.82%    | 0.415     | 0.02%   |
>     >|           | 1.0   | 96.09%    | 0.366     | 0.01%       |
>     >|           | 10.0  | 94.54%    | 0.336     | 0.00%   |
>
>     Our $\lambda_\text{der}$ ablation in Table C shows that better Jacobian alignment consistently leads to stronger attacks, while surrogate clean accuracy remains stable. This confirms that attack effectiveness depends on the gradient fidelity to the real pipeline, not on surrogate-specific artifacts.

---

> > ### Author Rebuttal · Reviewer_MiDQ · 2026-04-03
> >
> > The authors’ rebuttal is overall positive and has addressed part of my concerns. Regarding the surrogate training cost, the authors explained that the surrogate is not simply shifting the computational burden of an already feasible attack, but the current response still lacks clear validation of the actual time and resource overhead. Therefore, I consider W1 to be partially resolved. The response to W2 is more convincing than that to W1. The newly added Jacobian ablation provides effective evidence and has largely addressed my concern. Therefore, I am willing to keep my score unchanged.

---

> > > ### Author Response · Authors · 2026-04-06
> > >
> > > We thank the reviewer for the constructive feedback and for acknowledging that W2 has been largely addressed. We now provide the concrete time and resource measurements to fully address the remaining concern on W1.
> > >
> > > ----
> > >
> > > In tables F and G, we provide computational cost required for surrogate training and attack. All measurements are on a single NVIDIA A6000 (48GB).
> > > >**Table F. Computational cost for surrogate training on MNIST. We report wall-clock time and peak GPU memory for surrogate training.**
> > > >|Classifier|Hypernet|Naïve INR|FD INR |Naïve Emb.|FD Emb.|
> > > >|---|---|---|---|---|---|
> > > >|DWS|1.7h (10.2G)|1.7h (4.0G)|6.0h (4.0G)|-|-|
> > > >|Inr2Vec|2.2h (7.5G)|0.9h (0.5G)|1.1h (0.5G)|0.4h (0.3G)|1.5h (0.3G)|
> > > >|NFN|1.2h (7.4G)|1.7h (4.2G)|1.6h (4.2G)|-|-|
> > > >|NFT|13.7h (24.0G)|4.4h (22.4G)|21.7h (22.4G)|0.9h (22.1G)|3.8h (22.1G)|
> > > >|ScaleGMN|1.6h (7.6G)|1.6h (6.8G)|1.7h (6.8G)|-|-|
> > >
> > > >**Table G. Computational cost for surrogate attack on MNIST. We report wall-clock time and peak GPU memory for PGD attack with 20 steps.**
> > > >|Classifier|Hypernet|Naïve INR|FD INR|Naïve Emb.|FD Emb.|
> > > >|---|---|---|---|---|---|
> > > >|DWS|27min (10.0G)|24min (4.0G)|24min (4.0G)|-|-|
> > > >|Inr2Vec|5min (7.5G)|3min (0.3G)|3min (0.3G)|<1min (0.1G)|<1min (0.1G)|
> > > >|NFN|4min (7.3G)|2min (4.2G)|2min (4.2G)|-|-|
> > > >|NFT|24min (23.9G)|19min (22.4G)|19min (22.4G)|1min (1.2G)|1min (1.2G)|
> > > >|ScaleGMN|12min (7.6G)|8min (6.8G)|8min (6.8G)|-|-|
> > >
> > > While surrogate training introduces a one-time overhead, directly applying first-order attacks to INR pipelines is not tractable in practice due to the optimization-in-the-loop structure. The surrogate therefore enables practically feasible first-order attacks that would otherwise be difficult to realize. Moreover, the training cost is amortized -- once trained, the surrogate is reused across different attack algorithms and composite evaluation suites such as AutoAttack that sequentially apply multiple attacks to the same samples, with each additional evaluation costing only minutes.

---

### Official Review · Reviewer_nob8 · 2026-03-13

**Soundness:** 3
**Presentation:** 3
**Significance:** 3
**Originality:** 3
**Overall Recommendation:** 4
**Confidence:** 3

**Summary:**

The paper proposes a surrogate-based gradient approximation framework that replaces the non-differentiable INR fitting process with a differentiable surrogate model, enabling standard gradient-based adversarial attacks to systematically evaluate the robustness of INR-based classifiers. Two types of surrogate models are introduced. The first type, INR surrogates, directly predict the INR parameters from the input image. These include a hypernetwork based surrogate and simpler MLP-based surrogates that are trained to match the fitted INR weights. The second type, embedding surrogates, bypass both INR fitting and encoding by predicting the embedding representation used by the classifier. These models approximate the mapping from the input image to the classifier’s latent representation, enabling gradient-based attacks directly in embedding space.

**Compliance With Llm Reviewing Policy:**

Affirmed.

**Final Justification:**

My final recommendation is weak accept after rebuttal.

**Key Questions For Authors:**

How closely do the surrogate gradients approximate the true gradients of the INR fitting process?  for example gradient alignment or cosine similarity to better understand whether the attack strength depends on surrogate approximation quality.

**Limitations:**

Limitations are not extensively discussed.

**Strengths And Weaknesses:**

Strengths

1. This work studies the adversarial robustness of classifiers that operate on implicit neural representations (INRs) and also propose a surrogate-based robustness auditing framework. The main idea is to approximate the mapping from images to INR representations using differentiable surrogate models.

2. Experiments are conducted on MNIST, Fashion-MNIST, and CIFAR-10. The results show that although INR-based classifiers appear resistant to weak attacks such as random noise or transfer attacks, strong surrogate-based attacks reduce accuracy to nearly zero.

3. The authors also study factors affecting robustness, including INR architecture capacity, classifier design choices, and attack configurations. Finally, they evaluate several defenses such as diffusion denoising and adversarial training, finding that these methods provide limited improvements and often introduce accuracy trade-offs.


Weakness

1. All experiments are conducted on relatively small datasets (MNIST, Fashion-MNIST, CIFAR-10). Given that INRs are often used in higher-resolution tasks such as NeRF or large-scale image modeling, evaluations on larger datasets such as ImageNet would be relevant.

2. The surrogate attack framework mainly adapts existing surrogate-gradient ideas from adversarial domain, and the methodological contribution beyond applying them to INR pipelines is relatively modest, therefore theoretical insight into why INR representations become vulnerable under attacks would strength the work.

3. The adversarial training evaluation relies only on PGD-based attacks; however, robust AT standards typically require evaluation with stronger parameter-free attack suites such as AutoAttack and comparisons with additional INR-based baselines.

---

> ### Author Rebuttal · Authors · 2026-03-31
>
> **General Response** We thank you for your thoughtful feedback. We address each concern and provide new experimental results that strengthen our findings. Our key additions include:
> - Table A: **Extension to 3D point cloud classification on ShapeNet-10**
> - Table B: **Function-space AutoAttack** for AT evaluation
> - Table C: **FD supervision ablation** on gradient approximation quality
> - Table D: **Function-space Square Attack** as a gradient-free black-box baseline
> - Table E: **L2 regularization** as an INR-specific defense
>
> **W1. Limited dataset scale**
>
> We extend our evaluation to ShapeNet-10, a 3D point cloud classification benchmark used in Inr2Vec and NFN. We train five classifiers and corresponding INR surrogate models on ShapeNet-10, where Inr2Vec and NFN are trained on the same setting as their original paper. We also evaluate them under the same auditing framework.
>
> >**Table A. ShapeNet-10 results. We report accuracy in %. (ε=0.05, following Sun et al., 2021)**
> >| Classifier | Classifier Acc | Naïve Surr. Clean Acc. | Naïve Surr. Pert. Acc. | FD Surr. Clean Acc. | FD Surr Pert. Acc. |
> >|---|---|---|---|---|---|
> >|Inr2Vec|92.7|80.6|53.9|78.5|55.9|
> >|NFT|92.9|73.4|37.4|75.5|51.4|
> >|ScaleGMN|88.8|79.8|50.5|78.9|60.7|
> >|NFN|87.4|80.9|52.2|76.9|54.0|
> >|DWS|76.4|54.3|44.6|51.3|42.9|
>
> Our surrogate-based attacks reduce accuracy across all classifiers, confirming vulnerability of INR-based classifiers extends to 3D data. Clean accuracy gap between classifier and surrogate on ShapeNet-10 is greater than on 2D benchmarks due to the increased complexity of 3D INR fitting; since surrogate-based attacks can only underestimate vulnerability, these results represent a lower bound on the actual vulnerability.
>
> **W2. Theoretical insight into why INR representations become vulnerable**
>
> Our work provides the first systematic empirical analysis across multiple dimensions -- INR architectures, classifier designs, attack configurations, and attack surfaces -- to understand the source of vulnerability in INR-based classifiers. Through our extensive experiments, we have identified several factors that influence vulnerability:
> - **Expressivity increases vulnerability (Fig. 3):** Higher-capacity INRs are more susceptible, suggesting that the representation flexibility also amplifies sensitivity to adversarial perturbations.
> - **Function-space < Input-space attacks (Table 5):** Input-space attacks via surrogates are more effective than direct weight-space perturbations, indicating that the fitting process amplifies adversarial directions.
>
> **W3. AT evaluation with AutoAttack**
>
> Standard AutoAttack (Croce et al., 2020) operates in pixel space, incompatible with our function-space constraint. To address this, we develop AutoAttack-INR, adapting APGD-CE, APGD-DLR, and Square Attack to INR weight space (100 iterations, 5 restarts; FAB excluded due to incompatible minimum-norm formulation).
>
> >**Table B. AT evaluation with AutoAttack-INR on MNIST. ΔCln and ΔPrt(PGD) are from Table 7.**
> >|Model|ΔCln|ΔPrt(PGD)|ΔPrt(AutoAttack-INR)|
> >|---|---|---|---|
> >|Inr2Vec|−3.0|+35.6|0.0|
> >|DWS|+0.8|+16.1|+31.5|
> >|NFN|−1.5|+55.3|+39.2|
> >|NFT|−6.2|+69.8|+15.5|
> >|ScaleGMN|+0.1|+72.0|+8.6|
>
> DWS and NFN show genuine robustness gains, though reduced compared to PGD evaluation. Inr2Vec shows no improvement (0.0% both with and without AT).
>
> **Q1. Gradient approximation quality and its effect on attack strength**
>
> The FD surrogate is trained with the following loss:
> $$L = λ_{rec} · ‖\hat{\theta}(x) - \theta^*(x)‖² + λ_{cls} · CE(C(\hat{\theta}(x)), y) + λ_{der} · ‖(∂\hat{\theta}/∂x)·Δx − Δθ‖²,$$where the third term directly supervises the surrogate's Jacobian $∂\hat{\theta}/∂x$ to approximate the true Jacobian $∂\theta^\*/∂x$ of the INR fitting process.
>
> To show the attack strength with respect to the gradient approximation quality, we ablate $λ_{der}$ ∈ {0.01, 0.1, 1.0, 10.0} while fixing other parameters on MNIST to directly examine whether better gradient approximation leads to stronger attacks. Full results are provided in Table C (Response to Reviewer MiDQ).
>
> Since the third term in the training loss directly supervises gradient approximation, lower Jacobian loss indicates closer approximation to the true INR fitting gradient. Our experiment confirms this: surrogates with better gradient approximation consistently produce stronger attacks, with the tendency being more clear for classifiers not already at near-zero (NFT, ScaleGMN, DWS). This suggests that the near-zero adversarial accuracy achieved by our best surrogates reflects faithful gradient approximation.
>
> **References**
> * Andriushchenko, M. et al., Square Attack: a query-efficient black-box adversarial attack via random search. ECCV 2020.
> * Croce, F. et al., Reliable evaluation of adversarial robustness with an ensemble of diverse parameter-free attacks. ICML 2020.
> * Sun, J. et al., Adversarially robust 3D point cloud recognition using self-supervisions. NeurIPS 2021.

---

> > ### Author Rebuttal · Reviewer_nob8 · 2026-04-04
> >
> > Rebuttal have addressed my concerns. I will increase score accordingly.

---

### Decision · Program_Chairs · 2026-04-30

**Decision:**

Accept (regular)

**Comment:**

This paper systematically investigates the robustness of classification models based on Implicit Neural Representations (INR) under adversarial attacks, and proposes a surrogate-based attack evaluation framework that enables the application of first-order gradient attacks to the INR pipeline.
Overall, the majority of the reviewers acknowledge the technical soundness and the experimental results of the paper.
By employing a surrogate method, the paper reveals that the previously perceived "robustness" of INR models might actually stem from gradient obfuscation (or the unavailability of gradients); this insight is illuminating.
During the rebuttal phase, the authors adequately addressed the key concerns raised by the reviewers, including:Extending the evaluation to 3D data (ShapeNet-10) to broaden the experimental scope;Introducing AutoAttack-INR alongside additional attack methods;Analyzing the gradient approximation quality of the surrogate model (via Jacobian analysis);Providing supplementary data on the computational overhead for surrogate training and generating attacks;Exploring INR-specific defenses (e.g., L2 regularization).

These supplementary efforts were generally well-received by the reviewers. All reviewers explicitly stated that their concerns had been fully or largely resolved, and subsequently maintained or increased their scores.

Synthesizing the reviewers' feedback, I recommend accepting this paper.